# Stomatal control of leaf fluxes of carbonyl sulfide and $CO_2$ in a *Typha* freshwater marsh

Wu Sun[1], Kadmiel Maseyk[2,a], Céline Lett[3,a], and Ulli Seibt[1,a]

[1]Department of Atmospheric and Oceanic Sciences, University of California, Los Angeles, CA 90095-1565, USA
[2]School of Environment, Earth and Ecosystem Sciences, The Open University, Milton Keynes MK7 6AA, United Kingdom
[3]Laboratoire des Sciences du Climat et de l'Environnement, Université Paris Saclay, 91191 Gif-sur-Yvette, France
[a]formerly at Institute of Ecology and Environmental Sciences, Université Pierre et Marie Curie Paris 6, France

**Correspondence:** Wu Sun (wu.sun@ucla.edu) and Ulli Seibt (useibt@ucla.edu)

**Abstract.** Carbonyl sulfide (COS) is an emerging tracer to constrain land photosynthesis at canopy to global scales, because leaf COS and $CO_2$ uptake processes are linked through stomatal diffusion. The COS tracer approach requires knowledge of the concentration normalized ratio of COS uptake to photosynthesis, commonly known as the leaf relative uptake (LRU). LRU is known to increase under low light, but the environmental controls over LRU variability in the field are poorly understood due
to scant leaf scale observations.

Here we present the first direct observations of LRU responses to environmental variables in the field. We measured leaf COS and $CO_2$ fluxes at a freshwater marsh in summer 2013. Daytime leaf COS and $CO_2$ uptake showed similar peaks in the mid-morning and late afternoon separated by a prolonged midday depression, highlighting the common stomatal control on diffusion. At night, in contrast to $CO_2$, COS uptake continued, indicating partially open stomata. LRU ratios showed a clear
relationship with photosynthetically active radiation (PAR), converging to 1.0 at high PAR, while increasing sharply at low PAR. Daytime integrated LRU (calculated from daytime mean COS and $CO_2$ uptake) ranged from 1 to 1.5, with a mean of 1.2 across the campaign, significantly lower than previously reported laboratory mean value (~1.6). Our results indicate two major determinants of LRU—light and vapor deficit. Light is the primary driver of LRU because $CO_2$ assimilation capacity increases with light, while COS consumption capacity does not. Superimposed upon the light response is a secondary effect that high
vapor deficit further reduces LRU, causing LRU minima to occur in the afternoon, not at noon. The partial stomatal closure induced by high vapor deficit suppresses COS uptake more strongly than $CO_2$ uptake because stomatal resistance is a more dominant component in the total resistance of COS. Using stomatal conductance estimates, we show that LRU variability can be explained in terms of different patterns of stomatal vs. internal limitations on COS and $CO_2$ uptake. Our findings illustrate the stomata-driven coupling of COS and $CO_2$ uptake during the most photosynthetically active period in the field and provide
an in-situ characterization of LRU—a key parameter required for the use of COS as a photosynthetic tracer.

# 1 Introduction

Carbonyl sulfide (COS) is a tracer for land photosynthesis (Montzka et al., 2007; Campbell et al., 2008; Berry et al., 2013; Campbell et al., 2017). Globally, COS is mainly emitted from the ocean and anthropogenic activities and consumed by leaves and soils (Berry et al., 2013; Launois et al., 2015; Campbell et al., 2015; Whelan et al., 2017). Since ecosystem COS exchange is dominated by plant uptake (Berry et al., 2013), concurrent measurements of COS and $CO_2$ fluxes offer a way to separate photosynthesis and respiration from net carbon fluxes (e.g., Asaf et al., 2013; Billesbach et al., 2014). Understanding the relationship between leaf COS and $CO_2$ fluxes is therefore critical to COS-based estimates of canopy and regional photosynthesis.

In leaves, COS and $CO_2$ follow the same stomatal diffusional pathway and similar hydrolytic reactions catalyzed by carbonic anhydrase (CA), with the key difference being that the hydrolysis goes reversibly for $CO_2$ but one-way for COS (Protoschill-Krebs et al., 1996; Notni et al., 2007). The reaction of COS with CA yields $H_2S$ and $CO_2$ (Schenk et al., 2004; Notni et al., 2007), without any COS re-emission from leaves (Stimler et al., 2010). In contrast, $CO_2$ hydration is subject to chemical equilibrium that depends on its diffusional supply versus its demand from fixation, leading to retrodiffusion to the atmosphere. CA-mediated hydrolysis therefore serves as the sink reaction of COS in leaves, but not of $CO_2$.

The COS hydrolysis via CA is light independent (Goldan et al., 1988; Protoschill-Krebs et al., 1996) and efficient (Ogawa et al., 2013). Since the catalytic efficiency of CA in COS hydrolysis (Protoschill-Krebs et al., 1996; Ogée et al., 2016) is much higher than that of RuBisCO in $CO_2$ fixation (Tcherkez et al., 2006), COS is readily consumed within leaves and the hydrolysis is limited by COS supply (Goldan et al., 1988; Sandoval-Soto et al., 2005; Seibt et al., 2010; Stimler et al., 2010). Leaf COS uptake should therefore be mostly controlled by the sequence of conductances along the diffusional pathway and respond to environmental variables that regulate stomatal diffusion. It is well known that stomatal conductance responds to photosynthetically active radiation (PAR), because of the feedback from photosynthesis to stomatal conductance (Ball, 1988; Collatz et al., 1991), and to vapor deficit (Leuning, 1995), due to the optimization of water cost (Farquhar and Sharkey, 1982). Thus, through stomatal conductance, light and vapor deficit may control leaf COS uptake, even though COS hydrolysis itself depends on neither. In laboratory and field settings, light dependence of leaf COS uptake has been commonly observed (e.g., Stimler et al., 2011; Commane et al., 2015), but vapor deficit dependence has yet to be confirmed with observations.

At night, in contrast to the $CO_2$ emission, COS uptake may continue if stomata are not fully closed (Stimler et al., 2010). Constraining nighttime COS uptake is important for regional flux inversions (e.g., Berry et al., 2013; Hilton et al., 2017), because it may introduce biases when using large-scale COS drawdown patterns to infer changes in photosynthesis. Nighttime COS uptake has been observed in a wheat field (Maseyk et al., 2014), a boreal pine forest (Kooijmans et al., 2017), and temperate forests (Berkelhammer et al., 2014; Commane et al., 2015; Wehr et al., 2017). Most studies base their findings of nighttime COS uptake upon ecosystem scale observations, with only a handful of studies providing leaf-level evidence of nighttime COS uptake (Stimler et al., 2010; Berkelhammer et al., 2014; Kooijmans et al., 2017).

The relationship between leaf COS uptake and photosynthesis required for COS-based photosynthesis estimates is commonly expressed in a simple metric: leaf relative uptake (LRU). LRU is the ratio of leaf COS : $CO_2$ fluxes normalized by their respective ambient concentrations (Sandoval-Soto et al., 2005; Campbell et al., 2008). Laboratory studies have shown

that LRU varies with environmental conditions, especially PAR, and also by plant species (Stimler et al., 2010, 2011, 2012). In low light conditions, LRU decreases sharply with increasing PAR but becomes stable at PAR above ca. 500 µmol m$^{-2}$ s$^{-1}$ (Stimler et al., 2010, 2011). This LRU vs. PAR pattern is shared among many species despite interspecies variations of LRU values (Stimler et al., 2011). It results from the diverging responses of COS and $CO_2$ uptake in low light: $CO_2$ assimilation that is limited by both light and stomatal conductance decreases more rapidly than COS uptake that is controlled only by stomatal conductance. In addition, as COS uptake is more limited by stomatal conductance than $CO_2$ uptake due to the high efficiency of COS hydrolysis, high vapor deficit that triggers stomatal closure (also known as "midday depression") may have a stronger impact on COS uptake than on $CO_2$ uptake, and thus may lower LRU. In the field, the LRU–PAR relationship has only been approximated with ecosystem fluxes (Maseyk et al., 2014; Commane et al., 2015), not directly determined from leaf fluxes. The influence of vapor deficit on LRU has also not been studied. For COS-based canopy photosynthesis estimates, we need direct measurements of how LRU responds to PAR and vapor deficit in the field.

This study aims to characterize how light and vapor deficit drive variabilities in leaf COS uptake and LRU and to probe the stomatal mechanism behind LRU responses to these drivers. We hypothesize that (i) light dependence of instantaneous LRU is analogous to that reported in laboratory conditions, and this relationship is also preserved in daily integrated LRU; and (ii) high vapor deficit conditions reduce COS uptake more than $CO_2$ uptake and cause LRU to decrease. We report leaf COS and $CO_2$ fluxes measured in a *Typha latifolia* freshwater marsh during the peak growing season of June and July 2013. The *T. latifolia* at the site has high productivity and stomatal conductance (Tinoco Ojanguren and Goulden, 2013), which suits our study. We then examine how environmental variables control fluxes and LRU through stomatal mechanisms, and discuss the implications for COS-based photosynthesis estimates.

## 2 Methods

### 2.1 Site description

We measured leaf fluxes of COS, $CO_2$, and water from 31 May to 6 July 2013 (day of year 151–187) at the San Joaquin Freshwater Marsh (SJFM, 33°39′44.4″ N, 117°51′6.1″ W). The SJFM is located near the campus of the University of California, Irvine, at 3 m above sea level and 8 km northeast of the Pacific Ocean (Goulden et al., 2007). The SJFM is part of the University of California's Natural Reserve System. The site's history and management practices have been described in Goulden et al. (2007). Briefly, the SJFM is a mature freshwater marsh, the remnant of a once 2100 ha wetland along the San Diego Creek. Since the 1960s, the SJFM has been managed by flooding the area annually to a depth of approximately 1 m from December/January to March. The standing water recedes by evapotranspiration and subsurface drainage and eventually disappears by midsummer (Goulden et al., 2007). A flux tower (5 m tall) is located on a floating wooden platform near the northeastern edge of the SJFM. The platform is surrounded by dense vegetation dominated by *Typha latifolia* (broadleaf cattail). In contrast to upland species in a mediterranean climate that grow in the rainy winter or early spring, the growing season of the marsh plants is summer due to the standing water.

## 2.2 Experimental setup

Leaf fluxes of COS, $CO_2$, and $H_2O$ were measured with a flow-through (dynamic) chamber (Fig. 1a). The cylindrical chamber (18 cm diameter, 38 cm height, 10.3 L volume) consisted of PFA Teflon film stretched between two aluminum rings connected by rods. The PFA film was laid inside the structure such that only the film was in contact with the sampled air. The chamber enclosed the upper sections of six tall *T. latifolia* leaves of the same plant with an average width of 1.5 cm. The leaves extended above and below the chamber. The total leaf area in the chamber was estimated as 409.5 cm$^2$ by approximating the area of each leaf with a one-sided rectangle (length intersected by the chamber × width). Skirts of Teflon film were wrapped around the leaves to provide a seal at both ends of the chamber. Due to limitations on the sampling time of the COS analyzer, we did not install a replicate leaf chamber, but instead chose a high sampling frequency for the single leaf chamber.

Two fans were installed in the chamber for ventilation and mixing, respectively. On the inlet end, a high-speed axial fan (D344T, Micronel; 40×40 mm) provided ventilation to keep the chamber at ambient conditions (i.e., within 1 ppmv of ambient $CO_2$, tested at the start of the campaign). A second, smaller flat fan (F62, Micronel; 16 × 16 mm), attached to a stainless steel rod, was placed near the center of the chamber for air mixing. During the measurement period, the ventilation fan was turned off and its opening served as the inlet to allow airflow through the chamber. The mixing fan, in contrast, was kept running at all times.

The chamber was connected via a 0.25-inch PFA Teflon tubing to a Quantum Cascade Laser (QCL) analyzer (CW-QC-TILDAS, Aerodyne Research Inc., Billerica, MA, USA), with a 1 µm Teflon filter attached at the inlet of the analyzer. The analyzer was placed in an instrument enclosure on the platform. Flow through the analyzer was provided by a Varian TriScroll 600 pump (Agilent Technologies Inc., Santa Clara, CA, USA). Flow rate in the sampling tube was 6.4 standard liter per minute (slm), which corresponded to a chamber air turnover time of around 1.5 minutes. The pump was placed next to the nearest main power line near the entrance to the marsh site, and connected to the analyzer by a 150 m long 2-inch vacuum line. A solenoid valve at the inlet to the QCL was used to switch from the sampling line to a stream of dry $N_2$ (ultrahigh purity) for a one-minute background correction every hour. Data from the QCL analyzer were recorded at 10 Hz and stored on the QCL hard drive. The root-mean-square deviation of COS measurements at 10 Hz was 11–18 parts per trillion in volume (pptv).

Correction for water vapor effects on the dry mixing ratios of COS and $CO_2$ was done in the TDLWintel data acquisition software on the analyzer (Nelson, 2012). We did not use the same correction factors reported in Kooijmans et al. (2016) for the same make of QCL analyzer; however, a mock run of data processing with $CO_2$ concentration recalculated using their correction factor value resulted in a potential bias of only 0.12% ($r^2 = 0.999$). Thus, the flux uncertainty associated with the correction factor of water vapor effects was negligible (see the Supplement for details).

The leaf chamber was measured once per hour. Chamber operations were programmed on a CR1000 datalogger (Campbell Scientific, Inc., Logan, UT, USA). We monitored chamber air concentrations for a five-minute measurement period (i.e., while the ventilation fan was off), as well as the ambient air for one minute before and after measurement periods (i.e., while the ventilation fan was running). Leaf fluxes were calculated from the transient changes with respect to the interpolated inlet (ambient) concentrations (Fig. 1b). The apparent fluxes from the chamber material (PFA), characterized post hoc, were negligible—the

blank effects translated to apparent fluxes of $0.05 \pm 0.29$ pmol m$^{-2}$ s$^{-1}$ for COS and $0.02 \pm 0.15$ µmol m$^{-2}$ s$^{-1}$ for CO$_2$ when normalized against the leaf area (see the Supplement).

Various sensors were installed to record environmental data, including photosynthetically active radiation (PAR) (SQ-215, Apogee Instruments), ambient air temperature and humidity (HMP45AC, Vaisala), and chamber air and leaf temperature (type T thermocouples, PFA coated). These data were recorded at 10 s intervals on the CR1000 datalogger. Because of a wider gap in the canopy to the west of the chamber than to other directions, the chamber received slightly more light in the afternoon than in the morning. To account for the heterogeneity of the light microenvironment of the chamber, the PAR sensor was collocated with the chamber. All sensor data are released alongside the flux data (see Data Availability).

## 2.3 Calculation of leaf fluxes

A mass balance equation is formulated for the gas species being measured (COS, CO$_2$, or H$_2$O),

$$V \frac{dC}{dt} = q(C_a - C) + FA \tag{1}$$

where $C$ (mol m$^{-3}$) is the chamber headspace concentration of the gas, $C_a$ (mol m$^{-3}$) is the inlet (ambient) concentration, $q$ (m$^3$ s$^{-1}$) is the inlet flow rate, $V$ (m$^3$) and $A$ (m$^2$) are the chamber volume and leaf area, respectively, and $F$ (mol m$^{-2}$ s$^{-1}$) is the flux rate to be calculated. Solving the mass balance equation with the initial condition $C(t = 0) = C_a$, we obtain

$$C(t) = -\frac{FA}{q} \exp(-qt/V) + C_a + \frac{FA}{q} \tag{2}$$

The flux rate $F$ is

$$F = \frac{q}{A} \cdot \frac{C - C_a}{1 - \exp(-qt/V)} \tag{3}$$

Let $\hat{y} = C - C_a$ and $\hat{x} = \exp(-qt/V)$ be the variables for the regression, hence,

$$\hat{y} = \frac{FA}{q}(1 - \hat{x}) \tag{4}$$

The flux rate $F$ is then solved from the slope of the regression $\hat{y} \sim (1 - \hat{x})$. The standard error of the estimated $F$ is also obtained from the regression. The flux calculation method described above does not require a steady state to be reached in the chamber. A typical example of the chamber measurement period with the fitted curve of COS concentration changes is shown in Fig. 1b.

## 2.4 Data quality control

All leaf flux and meteorological data have been quality checked and filtered. Conspicuously unrealistic data points in the meteorological data were removed. For the flux data, we used several independent criteria to filter measurements. First, measurement periods with serious misfit of the shape of concentration changes during chamber closure or with strong drift in the ambient concentrations were discarded. Second, flux estimates associated with large root-mean-square errors between fitted and observed concentrations were also discarded. Next, outliers in flux data were detected using the Tukey's interquartile range

method (Wilks, 2011). In addition, strongly positive $CO_2$ fluxes during the day and strongly negative $CO_2$ fluxes at night were also removed. Only the data points that passed all these filtering criteria were kept in the final data for analysis. After the filtering, 73.9% of COS flux observations and 54.3% of $CO_2$ flux observations were retained.

## 2.5 Calculation of flux-derived variables

### 2.5.1 Stomatal conductance of water and total conductances of $CO_2$ and COS

Stomatal conductance of water ($g_{s, H_2O}$, mol m$^{-2}$ s$^{-1}$) is calculated from water flux measurements,

$$g_{s, H_2O} = \frac{F_{H_2O}}{D} \tag{5}$$

where $F_{H_2O}$ is the water flux (mmol m$^{-2}$ s$^{-1}$), $D$ is the leaf-to-air water vapor deficit expressed in mole fraction (mmol mol$^{-1}$). The mole-fraction vapor deficit $D$ is calculated from

$$D = \frac{e_{sat}(T_{leaf})}{p} - \chi_{H_2O} \tag{6}$$

where $e_{sat}$ (Pa) is the saturation water vapor pressure as a function of temperature (Goff and Gratch, 1946), $T_{leaf}$ (°C) is the leaf temperature (see the Supplement for details), $p$ (Pa) is the ambient pressure, and $\chi_{H_2O}$ (mmol mol$^{-1}$) is the water vapor mixing ratio in the chamber air.

The total conductances of COS ($g_{tot, COS}$, mol m$^{-2}$ s$^{-1}$) and $CO_2$ ($g_{tot, CO_2}$, mol m$^{-2}$ s$^{-1}$) are calculated from:

$$g_{tot, COS} = -\frac{F_{COS}}{\chi_{COS}} \tag{7}$$

$$g_{tot, CO_2} = -\frac{F_{CO_2}}{\chi_{CO_2}} \tag{8}$$

where $F_{COS}$ (pmol m$^{-2}$ s$^{-1}$) and $F_{CO_2}$ (µmol m$^{-2}$ s$^{-1}$) are leaf COS and $CO_2$ fluxes, $\chi_{COS}$ (pmol mol$^{-1}$) and $\chi_{CO_2}$ (µmol mol$^{-1}$) are mixing ratios of COS and $CO_2$ in the chamber air, respectively. Note that the intercellular concentrations of COS and $CO_2$ are canceled out from these equations by approximating their biochemical reaction rates with hypothetical (but mathematically convenient) 'biochemical conductances' (Stimler et al., 2010; Berry et al., 2013), which are then included in the total conductances.

### 2.5.2 Instantaneous and time-integrated leaf relative uptake ratios

Instantaneous leaf COS : $CO_2$ relative uptake (LRU) is defined as the ratio of COS and $CO_2$ fluxes normalized by their respective mixing ratios (Sandoval-Soto et al., 2005; Campbell et al., 2008; Whelan et al., 2017),

$$LRU = \frac{F_{COS}}{F_{CO_2}} \cdot \frac{\chi_{CO_2}}{\chi_{COS}}, \text{ where } F_{COS} < 0 \text{ and } F_{CO_2} < 0 \tag{9}$$

LRU is a dimensionless quantity. We confine our LRU analysis to occasions where both COS and $CO_2$ fluxes are negative (i.e., showing net uptake). Hence, LRU is only calculated during the daytime and is always positive.

We also calculate the all-day mean LRU ($\text{LRU}_{\text{all-day}}$) and the daytime mean LRU ($\text{LRU}_{\text{daytime}}$) of each day using

$$\text{LRU}_{\text{all-day}} = \frac{\left(\sum\limits_{i=0}^{23} F_{\text{COS}}^i\right) \cdot \left(\sum\limits_{i=0}^{23} \chi_{\text{CO}_2}^i\right)}{\left(\sum\limits_{i=0}^{23} F_{\text{CO}_2}^i\right) \cdot \left(\sum\limits_{i=0}^{23} \chi_{\text{COS}}^i\right)} \tag{10}$$

$$\text{LRU}_{\text{daytime}} = \frac{\left(\sum\limits_{i=6}^{19} F_{\text{COS}}^i\right) \cdot \left(\sum\limits_{i=6}^{19} \chi_{\text{CO}_2}^i\right)}{\left(\sum\limits_{i=6}^{19} F_{\text{CO}_2}^i\right) \cdot \left(\sum\limits_{i=6}^{19} \chi_{\text{COS}}^i\right)} \tag{11}$$

where $i$ is the truncated hour number (integer), in local daylight-saving time (UTC–7). The daytime period is determined with solar elevation angle $> 0°$, which translates roughly to between 06:00 and 20:00. In each period of calculation, missing data points are gap-filled with the mean in that period.

### 2.5.3 Contributions of stomatal component to the total resistance

To assess the relative importance of the stomatal limitation on COS and $CO_2$ uptake with respect to internal limitations (mesophyll conductance and biochemical reactions), we calculate the ratios of stomatal resistance to total resistance for COS ($r_{\text{COS}}^*$) and $CO_2$ ($r_{\text{CO}_2}^*$),

$$r_{\text{COS}}^* = \frac{r_{\text{s, COS}}}{r_{\text{tot, COS}}} = \frac{g_{\text{tot, COS}}}{g_{\text{s, COS}}} = \frac{g_{\text{tot, COS}}}{g_{\text{s, H}_2\text{O}}/2.01} \tag{12}$$

$$r_{\text{CO}_2}^* = \frac{r_{\text{s, CO}_2}}{r_{\text{tot, CO}_2}} = \frac{g_{\text{tot, CO}_2}}{g_{\text{s, CO}_2}} = \frac{g_{\text{tot, CO}_2}}{g_{\text{s, H}_2\text{O}}/1.66} \tag{13}$$

where 2.01 is the water-to-COS ratio of diffusivity in air, and 1.66 is the water-to-$CO_2$ ratio of diffusivity in air (Seibt et al., 2010). The reason to switch from conductance to its reciprocal—resistance—is simply that different resistance components are *additive*.

### 2.6 Fitting light response curves for leaf COS and $CO_2$ fluxes and LRU

We used the LOWESS (locally weighted scatterplot smoothing) regression method to obtain smooth light response curves for COS flux, $CO_2$ flux, and LRU (see Fig. 5). The LOWESS regression method is a nonparametric method that does not require any a priori known relationship between the predictor (here, PAR) and the response variables (COS flux, $CO_2$ flux, and LRU). At each point in the range of the predictor, a low-degree polynomial is fitted to all the neighboring points to estimate the least squares response, weighted by the distances between the neighboring points and the current point (Cleveland et al., 1992). The calculation was performed with the Python statsmodels package, version 0.8.0 (Seabold and Perktold, 2010).

## 3   Results

### 3.1   Leaf fluxes of COS, $CO_2$, and water

During the campaign period in summer 2013 covering the peak growing season of *Typha latifolia*, meteorological conditions changed little except for a few cloudy days (8, 9, and 30 June 2013 in Fig. 2d), and the diurnal patterns of leaf COS, $CO_2$, and
5   $H_2O$ fluxes therefore also remained similar (Fig. 2a–c). The diurnal patterns of leaf fluxes and related variables are visualized with hourly binned medians and quartiles (Fig. 3).

In the daytime, leaf uptake of COS and $CO_2$ showed similar patterns (Fig. 3a, b), with uptake peaks in the morning and afternoon separated by a prolonged midday depression around local noon (13:00). The midday depression was up to 36% for COS (5.5 pmol $m^{-2}$ $s^{-1}$ at 14 h versus 8.5 pmol $m^{-2}$ $s^{-1}$ at 11 h) and 40% for $CO_2$ (3.7 μmol $m^{-2}$ $s^{-1}$ at 13 h versus 6.1
10   μmol $m^{-2}$ $s^{-1}$ at 17 h), respectively. The morning peaks coincided for the two fluxes at around 11:00, whereas the afternoon peak occurred a bit later for COS (18:00) than for $CO_2$ (17:00). The afternoon peak of $CO_2$ flux was slightly stronger than its morning peak (Fig. 3b), probably because the chamber received slightly more light in the afternoon than in the morning (Fig. 3e). Leaf transpiration showed a decline at 11:00 (Fig. 3c), but with an earlier afternoon peak (16:00) that coincided with the maximum vapor deficit (Fig. 3f). Contrary to COS and $CO_2$ fluxes, the diurnal pattern of water flux was strongly
asymmetric due to the high vapor deficit in the afternoon (Fig. 3f).

In contrast to daytime fluxes, nighttime fluxes of COS and $CO_2$ showed diverging patterns. At night, $CO_2$ was emitted from leaf respiration (Fig. 3b), whereas COS uptake continued (Fig. 3a). Both fluxes had significantly smaller magnitudes than during the day, with $CO_2$ emissions of around 1 μmol $m^{-2}$ $s^{-1}$, and COS uptake of around 2–3 pmol $m^{-2}$ $s^{-1}$. Note that although COS emissions were occasionally observed at night (Fig. 2a), they were likely caused by random error due to high
flow rates (~6 slm), and the hourly medians indeed showed a robust pattern of nighttime COS uptake (Fig. 3a). When averaged over the whole campaign, nighttime COS uptake was 23% of the total daily COS uptake by leaves. Nighttime transpiration was minimal (Fig. 3c) as the vapor deficit was close to zero at night (Fig. 3f).

COS flux was linearly correlated with $CO_2$ flux (Fig. 4a), with $r^2 = 0.49$ ($p = 7.6 \times 10^{-64}$). The relationship between COS and water fluxes was nonlinear (Fig. 4b) and showed a wide spread in the daytime due to the asymmetric diurnal pattern of
water fluxes (Fig. 3c). As a result, the correlation between them was lower (Fig. 4b), showing an $r^2$ of 0.32 ($p = 4.7 \times 10^{-57}$). The unbiased distance correlation (dCor; Székely et al., 2007) was also calculated as a more robust measure for the nonlinear correlation between COS and water fluxes, and $dCor^2 = 0.37$. At night, COS fluxes showed stronger variability than water fluxes because vapor deficit that drives transpiration was small (Fig. 3f).

The midday depression was also evident in the light responses of fluxes. Both COS and $CO_2$ uptake rates increased with
PAR until they became light saturated, and then decreased at high light and high vapor deficit (Fig. 5a, b). According to the smoothed light response curves, at a typical midday light level (1800 μmol $m^{-2}$ $s^{-1}$), COS uptake drops by 37% from the peak value of 7.5 pmol $m^{-2}$ $s^{-1}$ (at PAR = 493 μmol $m^{-2}$ $s^{-1}$) to 4.7 pmol $m^{-2}$ $s^{-1}$, while $CO_2$ uptake drops by 31% from the peak value of 5.3 μmol $m^{-2}$ $s^{-1}$ (at PAR = 740 μmol $m^{-2}$ $s^{-1}$) to 3.7 pmol $m^{-2}$ $s^{-1}$.

## 3.2 Diurnal patterns of stomatal conductance and total conductance

Stomatal conductance ($g_{s, H_2O}$) derived from water measurements showed a distinct period of midday depression in its diurnal pattern (Fig. 6a). $g_{s, H_2O}$ was the highest in the early morning after daybreak, but started to drop quickly as the vapor deficit picked up, reaching its minimum at local noon (13:00). In the late afternoon, stomatal conductance slowly rebounded and remained relatively stable, but was still lower than the early morning level. Nighttime stomatal conductance was unable to be estimated from water measurements due to large uncertainty introduced by low vapor deficit and water flux.

The total conductance of COS ($g_{tot, COS}$) exhibited broadly similar diurnal pattern to that of $g_{s, H_2O}$, but lagged by 1 hour (Fig. 6a). A midday depression period was also visible in the diurnal trend of $g_{tot, COS}$. At night, $g_{tot, COS}$ remained at a stable, low level.

The ratios of stomatal resistance to total resistance of COS ($r^*_{COS}$) and of $CO_2$ ($r^*_{CO_2}$) indicated that stomatal limitation was the dominant component in the diffusional pathways of both gases during most of the daytime (Fig. 6b). Despite large uncertainties associated with these ratios, $r^*_{COS}$ was higher than $r^*_{CO_2}$ by 20–40% around midday (10:00–13:00) at a significance level of $p < 0.05$ (paired two-sample $t$-tests), indicating stronger stomatal limitation on COS uptake. However, in the late afternoon (15:00–17:00) the difference between stomatal limitations on COS uptake and on $CO_2$ uptake was small and statistically insignificant (Fig. 6b).

## 3.3 Leaf relative uptake ratios

The instantaneous leaf relative uptake (LRU) showed an asymmetric U-shape diurnal pattern (Fig. 3d). LRU had highest values of 2–3 (medians binned by the hour) near dawn or dusk, with a gradual decrease throughout the morning and early afternoon, and then had minima around 0.9 at 15:00.

The diurnal pattern of LRU (Fig. 3d) was consistent with the LRU response to PAR (Fig. 5c). With increasing PAR, LRU decreased to around 1.0 at PAR above 500–600 $\mu mol\ m^{-2}\ s^{-1}$ (Fig. 5c). Surprisingly, the lowest LRU values during the day did not occur at the time of the highest PAR (Fig. 3d), but rather at the time of the highest vapor deficit (Fig. 3f) and moderately strong PAR (1000–1400 $\mu mol\ m^{-2}\ s^{-1}$) due to the stronger stomatal limitation on fluxes as a response to the high vapor deficit. The timing of the lowest LRU (Fig. 3d), around 15:00, coincided with the timing of the highest vapor deficit.

The all-day mean LRU at this site showed large day-to-day variations (1.4–3.6) and also had large uncertainty due to the random error in nighttime $CO_2$ fluxes (Fig. 7a). In contrast, the daytime mean LRU, averaged over the daylight period of 14 hours, did not show strong variability (1.0–1.8) and had an average value of 1.2 across the campaign. The daytime mean LRU was consistently lower than the all-day mean LRU, since the latter included nighttime COS uptake and $CO_2$ emissions (Fig. 7a). Following Maseyk et al. (2014), a power law relationship was fitted between daytime mean LRU and daytime mean PAR: LRU $= a \cdot PAR^b$ (or rather, a linear model between ln LRU and ln PAR), which yielded $a = 24.0689$ and $b = -0.4620$, with $r^2 = 0.28$ and $p = 0.012$ (Fig. 7b). On overcast days, the daytime mean LRU values were higher than on clear days (Fig. 7a), as is expected from the light response of LRU.

## 4 Discussion

### 4.1 Competition between stomatal and internal limitations underlie the responses of leaf relative uptake to light and vapor deficit

We have reaffirmed in field conditions that LRU decreases with increasing PAR (Fig. 5c), consistent with laboratory studies
(Stimler et al., 2010, 2011). The large sample size from high frequency measurements supported a robust analysis of LRU variability despite experimental limitations. Thanks to a strong diurnal variation of vapor deficit in this ecosystem, we were able to identify a further reduction in LRU caused by high vapor deficit—a secondary effect superimposed on the light dependence of LRU. But how are stomata responsible for the observed LRU responses?

Using the ratio of stomatal resistance to total resistance as a metric of the relative importance of stomatal limitation (Fig. 6b),
we can recognize how the dynamics of stomatal vs. internal limitations regulate LRU. At the leaf scale, LRU manifests the ratio between the stomatal limitation on COS uptake ($r_{COS}^*$) and that on $CO_2$ uptake ($r_{CO_2}^*$) (compare Eqs. 12 and 13 to Eq. 9):

$$\text{LRU} \equiv \frac{g_{\text{tot, COS}}}{g_{\text{tot, CO}_2}} = \frac{0.83 \cdot r_{COS}^*}{r_{CO_2}^*} \tag{14}$$

where 0.83 is the COS-to-$CO_2$ ratio of diffusivity in air (Seibt et al., 2010). The equation shows that LRU becomes smaller when $r_{COS}^*$ and $r_{CO_2}^*$ get closer, providing a simple mechanistic interpretation of LRU variability.
The light response of LRU arises from the fact that with respect to the same increase of PAR, the relative increase of COS uptake is less than that of $CO_2$ uptake (Fig. 5a, b), i.e.,

$$\frac{\partial \text{LRU}}{\partial \text{PAR}} < 0 \iff \frac{1}{|F_{COS}|} \frac{\partial |F_{COS}|}{\partial \text{PAR}} < \frac{1}{|F_{CO_2}|} \frac{\partial |F_{CO_2}|}{\partial \text{PAR}}$$

$$\left( F_{COS} < 0 \text{ and } F_{CO_2} < 0 \right) \tag{15}$$

Increasing PAR drives an increase in $CO_2$ assimilation rates, which in turn leads to an increase in stomatal conductance to facilitate optimal $CO_2$ uptake. This increase in stomatal conductance also enables higher COS uptake rates, but as COS
hydrolysis is light independent (Protoschill-Krebs et al., 1996), there is a proportionally less increase in COS than $CO_2$ uptake. In other words, with the increase of PAR, both stomatal and biochemical limitations for $CO_2$ assimilation are relaxed, whereas for COS only the stomatal limitation is relaxed. This explanation is supported by indirect evidence in $r_{COS}^*$ and $r_{CO_2}^*$: from 06:00 to 13:00 there was a higher relative increase of $r_{CO_2}^*$ than that of $r_{COS}^*$ (Fig. 6b), which means the reduction of non-stomatal limitation—attributed mainly to the increases in biochemical reaction rates—is higher for $CO_2$ than for COS.
Stomatal response to vapor deficit, such as the midday depression (Fig. 6a), is a well-known behavior that serves to optimize water cost against carbon gain (e.g., Tenhunen et al., 1984; Collatz et al., 1991). However, the fact that vapor deficit has differential effects on COS and $CO_2$ uptake appears puzzling, since it does not affect COS and $CO_2$ biochemical reactions, and nor is it known to affect mesophyll conductance. A closer scrutiny of the stomatal limitations of COS and $CO_2$ (Fig. 6b) shows that the difference between $r_{COS}^*$ and $r_{CO_2}^*$ became smaller during the period of peak vapor deficit (14:00–17:00). Although
vapor deficit has the same effect on $g_{\text{s, COS}}$ and $g_{\text{s, CO}_2}$, it can change the proportion of stomatal vs. internal components in the

total resistance to the uptake, because COS uptake is always more stomatal-conductance-limited than $CO_2$ uptake ($r^*_{COS}$ always higher than $r^*_{CO_2}$ in Fig. 6b)—a direct consequence of the higher catalytic efficiency of CA than RuBisCO. Thus, vapor deficit controls LRU variability, but is less influential than PAR.

Since the mesophyll conductance is also a component in the internal conductance, it is worthy of note that the increase of mesophyll conductance with leaf temperature (Bernacchi, 2002) may have contributed to the dynamics of stomatal vs. internal limitations over the course of the daytime, as is shown in Wehr et al. (2017), although we lack relevant data to separate biochemical limitation from mesophyll limitation.

## 4.2 Nighttime COS uptake is a significant portion of COS budget

During this campaign, nighttime uptake contributed to 23% of the total daily leaf COS uptake. This fraction is comparable to those reported from a wheat field ($29 \pm 5\%$, Maseyk et al., 2014), an alpine temperate forest (25–30%, Berkelhammer et al., 2014), a boreal pine forest (17%, Kooijmans et al., 2017), and a New England mixed forest (< 20% after subtracting soil uptake, Commane et al., 2015; Wehr et al., 2017). Collectively, these studies indicate that nighttime uptake is typically 17–30% of the total canopy COS budget, a fraction too large to be ignored in ecosystem or regional COS budget. Understanding nighttime COS uptake is necessary for the success of COS-based photosynthesis estimates on daily and longer timescales.

The *T. latifolia* leaves showed a mean value of 5.0 mmol m$^{-2}$ s$^{-1}$ for the total conductance of COS ($g_{tot, COS}$) at night (Fig. 6a). Assuming that the internal conductance of COS at night is the same as its daytime average, we obtain an estimate of nighttime $g_{s, COS}$, 6.4 mmol m$^{-2}$ s$^{-1}$ (see the Supplement for detailed calculations). This estimate of the nighttime $g_{s, COS}$ is at the lower end of values reported from other ecosystems: 1.6 mmol m$^{-2}$ s$^{-1}$ for a New England mixed forest (Wehr et al., 2017), 5–30 mmol m$^{-2}$ s$^{-1}$ for a Scots pine forest (Kooijmans et al., 2017), 11.5 mmol m$^{-2}$ s$^{-1}$ for a wheat field (Maseyk et al., 2014), and 13–20 and 22–66 mmol m$^{-2}$ s$^{-1}$ for pine and poplar trees, respectively, in an alpine temperate forest (Berkelhammer et al., 2014). The nighttime stomatal conductance shows a large variability among different species.

In land biosphere models, nighttime stomatal conductance is often a fixed value regardless of plant type and water status, e.g., $g_{s, H_2O} = 10$ mmol m$^{-2}$ s$^{-1}$ in the Community Land Model v4.5 (Oleson et al., 2013). The fixed-value parameterization may introduce biases to the nighttime COS fluxes and long-term COS budget in regional simulations, which may in turn propagate into the COS-based photosynthesis estimates. To constrain nighttime COS uptake requires an understanding of the variability of nighttime stomatal conductance among plant species and ecosystem types. Water and COS flux measurements need to be used in conjunction to derive robust estimates of nighttime stomatal conductance. We expect COS measurements to be particularly useful for stomatal conductance estimates in tropical rainforests and other environments that experience high humidity conditions, provided that the variability of the internal conductance of COS is well understood.

## 4.3 Implications on COS-based photosynthesis estimation

LRU is an important empirical parameter used to derive ecosystem photosynthesis (also known as gross primary productivity, GPP) from COS measurements on spatial scales ranging from the ecosystem to the continent (Asaf et al., 2013; Commane et al., 2015; Hilton et al., 2015). Choosing a representative LRU for COS-based GPP estimation is crucial and challenging.

In addition to its environmental controls, LRU also varies among plant species (Stimler et al., 2012). For the *T. latifolia*, the asymptotic LRU value at high light (PAR > 600 µmol m$^{-2}$ s$^{-1}$) is around 1.0 (Fig. 5c). This value is much lower than the mean LRU of $1.61 \pm 0.26$ from laboratory measurements across a range of species (Stimler et al., 2012), which has been used as a representative LRU in ecosystem-scale (e.g., Asaf et al., 2013) and regional-scale GPP inversion studies (e.g., Hilton et al., 2015). The low asymptotic LRU of *T. latifolia* is, however, not surprising according to the mechanistic LRU model in Seibt et al. (2010), which describes that LRU is positively related to the ratio of intercellular $CO_2$ to the ambient $CO_2$ ($C_i/C_a$). As *T. latifolia* often has a high photosynthetic capacity (e.g., Tinoco Ojanguren and Goulden, 2013; Jespersen et al., 2017), its $C_i/C_a$ ratio may be lower than other species, thus contributing to the low LRU. Additionally, it has been noted that the aerenchyma of *T. latifolia* serves as a conduit to transport reduced gases from the rhizosphere to the atmosphere (Bendix et al., 1994; Yavitt and Knapp, 1998), which may act as a hidden COS source. Although the presence of this mechanism cannot be ruled out with our method, as it is an intrinsic process of the marsh plant and part of the plant–atmosphere COS exchange, and therefore the LRU measured here remains relevant for larger scale applications in this, and similar, ecosystems. Relatively low LRU values have also been reported from other ecosystems, for example, 1.3 in a wheat field (Maseyk et al., 2014) and 1.2 in a mixed temperate forest at high PAR (Commane et al., 2015). This suggests that for the success of COS-based GPP estimation, LRU needs to be locally constrained on the dominant species in an ecosystem, rather than assumed to be a constant.

For regional scale applications, the time-integrated LRU can be more relevant than the instantaneous LRU. Large scale patterns of COS and $CO_2$ drawdown imprinted in an air parcel are spatiotemporally integrated features, because the transport of surface uptake signals to the planetary boundary layer takes time and may be affected by the entrainment with other parcels along the way. Our results of time-integrated LRU show that although daytime mean LRU and PAR are correlated, nighttime leaf respiration and COS uptake create large variability in the all-day mean LRU, which decouples it from PAR (Fig. 7b). This suggests that a bottom-up scaling is unlikely to offer reliable daily LRU values for regional scale applications. Instead, LRU that is diagnostically calculated from biosphere models such as the Simple Biosphere model (Berry et al., 2013; Hilton et al., 2015) would be more appropriate for COS–GPP inversion studies, provided that model parameterizations are validated against observations.

## 5   Conclusions

Our field study has shown that leaf COS and $CO_2$ fluxes share similar diurnal patterns driven by the common stomatal responses to light and vapor deficit, showing dual peaks of uptake separated by a prolonged midday depression period. We have validated the light dependence of LRU directly at the leaf level in field conditions. LRU converges to around 1.0 at light-saturated conditions for *Typha latifolia*, much lower than many other species due possibly to its high photosynthetic capacity. In addition to light, vapor deficit is identified as a secondary driver of LRU, acting to reduce LRU further in the afternoon (15:00–17:00) from its light-saturated value.

Stomatal conductance derived from water measurements has provided process-level insights into the diurnal variability of LRU. Since the biochemical sink of COS is light independent, COS uptake is less reaction-limited compared with $CO_2$ uptake.

With increasing light, the assimilation capacity for $CO_2$ increases but is unchanged for COS, causing LRU to decrease regardless of the stomatal coupling between COS and $CO_2$. The reduction in stomatal conductance induced by high vapor deficit affects COS uptake more than $CO_2$ uptake, since COS uptake is more stomatal-conductance-limited, causing a further reduction in LRU. In summary, LRU variability is regulated by the relative influences of stomatal limitation vs. internal limitation on COS and $CO_2$ uptake.

The coupling between leaf COS and $CO_2$ fluxes and the predictability of LRU lend strong support to the use of COS as a quantitative tracer of canopy photosynthesis. More unknowns exist in the process-level controls of LRU, especially the variability of internal conductance. We expect that future studies may find the use of LRU as a diagnostic of stomatal processes to be interesting.

*Data availability.* Data presented here can be found in the University of California Curation Center (UC3) Merritt data repository at https://doi.org/10.15146/R37T00.

*Author contributions.* U.S. designed and supervised the research. All authors conducted the fieldwork. W.S. and U.S. performed data analysis. W.S., U.S., and K.M. wrote the paper with contributions from all co-authors.

*Competing interests.* The authors declare no conflict of interest.

*Acknowledgements.* The work was performed at the San Joaquin Freshwater Marsh (SJFM) Reserve of the University of California Natural Reserve System. We thank Mike Goulden at UC Irvine for help and discussions, and Bill Bretz and Peter Bowler for assistance at the SJFM UC Reserve. This work was supported by the European Research Council (ERC) Starting Grant no. 202835 and NSF CAREER Award no. 1455381 to U.S. We thank Teresa Gimeno, Mary Whelan, and an anonymous reviewer for their time and effort that greatly helped to improve this manuscript.

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

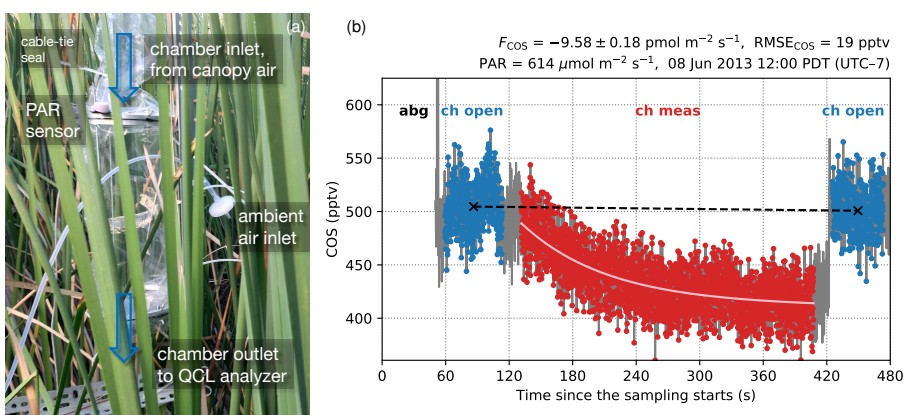

**Figure 1.** (a) A schematic diagram of the leaf chamber. (b) A typical sampling period on the leaf chamber illustrated with COS concentration measurements. The first minute is for auto-background spectral correction (abg) using $N_2$ gas. The sampling system then switches to the chamber line with the ventilation fan turned on (ch open) for one minute. Then the ventilation fan is turned off for five minutes to measure flux signals in the chamber (ch meas), and after that is turned on again for one minute (ch open). The fitted curve for concentration changes is shown in light pink. The black dashed line represents the zero-flux baseline correction to account for the drift in the measured ambient concentrations.

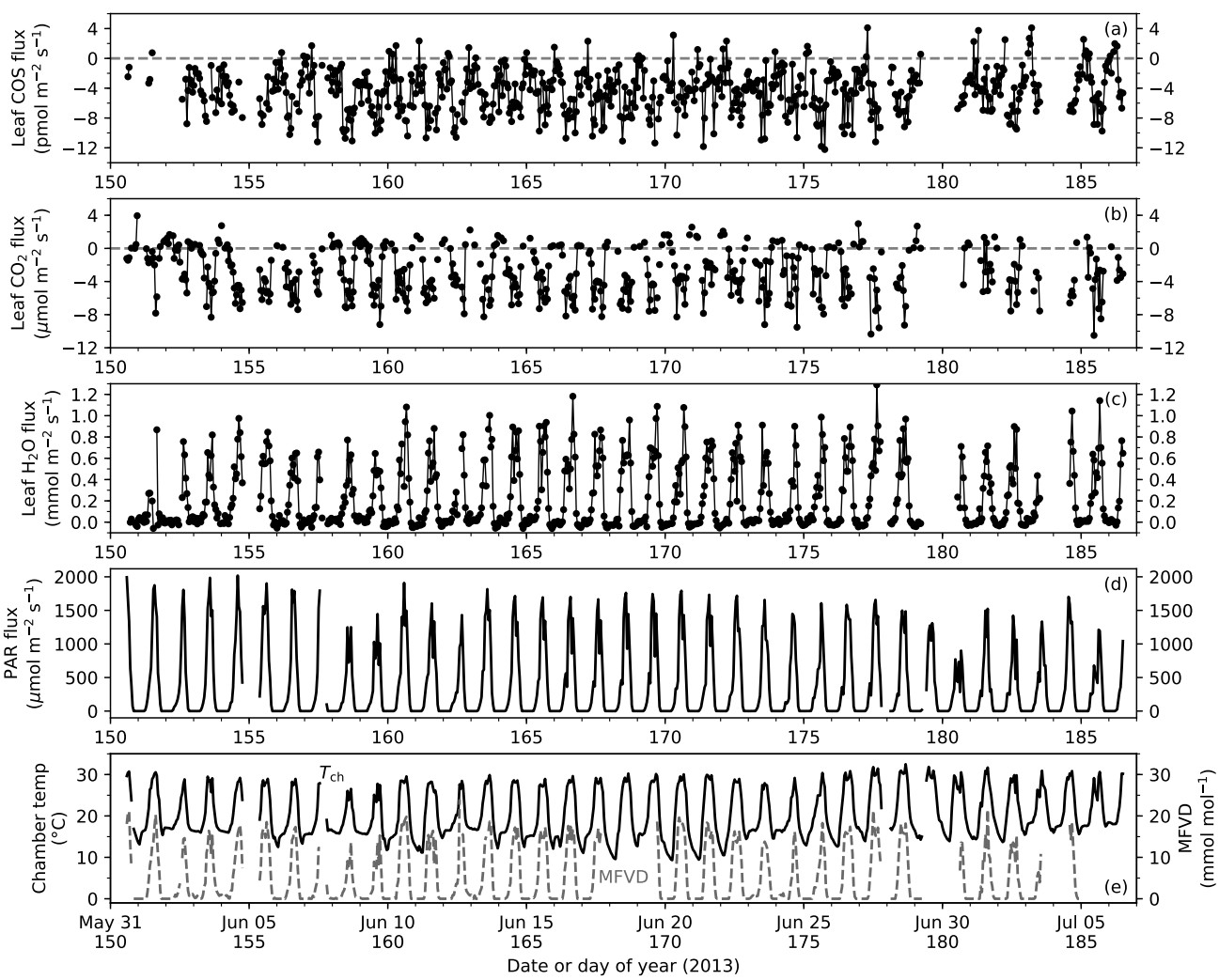

**Figure 2.** Time series of leaf COS (a), $CO_2$ (b) and water (c) fluxes, photosynthetically active radiation (PAR) at the leaf chamber (d), chamber air temperature (e, black solid line; $T_{ch}$) and leaf-to-air vapor deficit in mole fraction (e, gray dashed line; MFVD). Ticks on *x*-axes indicate the starts of the days (0000 h).

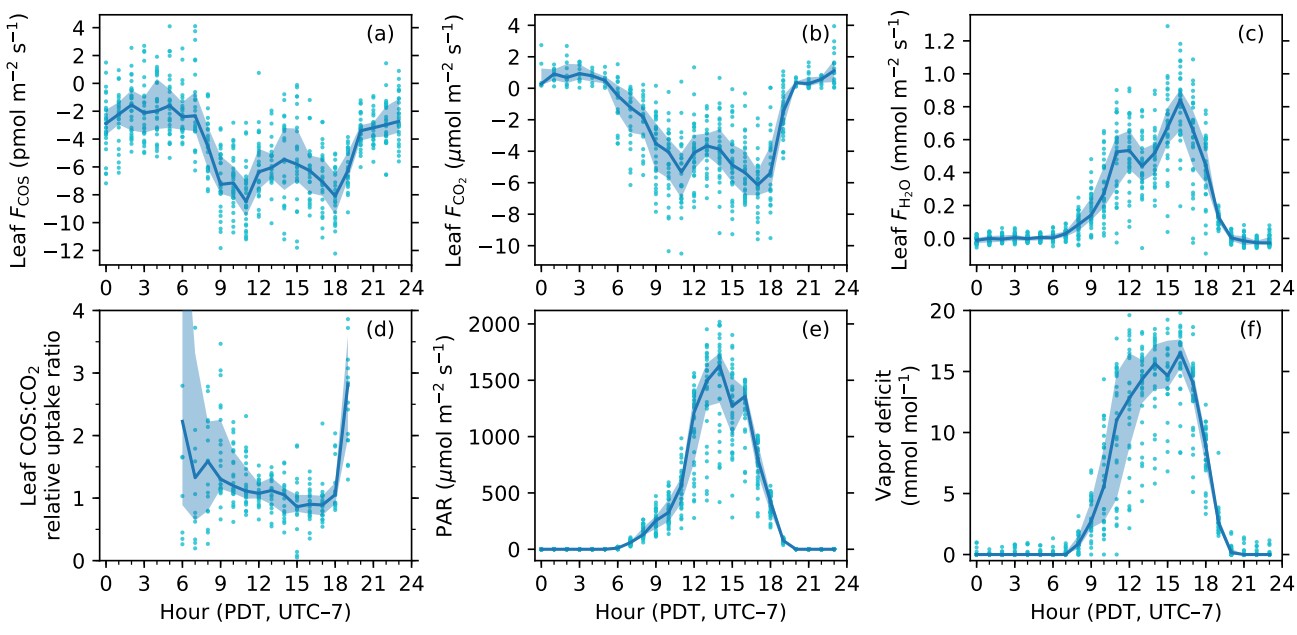

**Figure 3.** Diurnal patterns of leaf COS (a), $CO_2$ (b) and water (c) fluxes, leaf relative uptake ratio (d), PAR at the leaf chamber (e), and leaf-to-air vapor deficit in mole fraction (f). The solid curves show medians binned by the hour of the day (Pacific Daylight Time, UTC–7), and the upper and lower bounds of shaded areas are 25th and 75th percentiles, respectively.

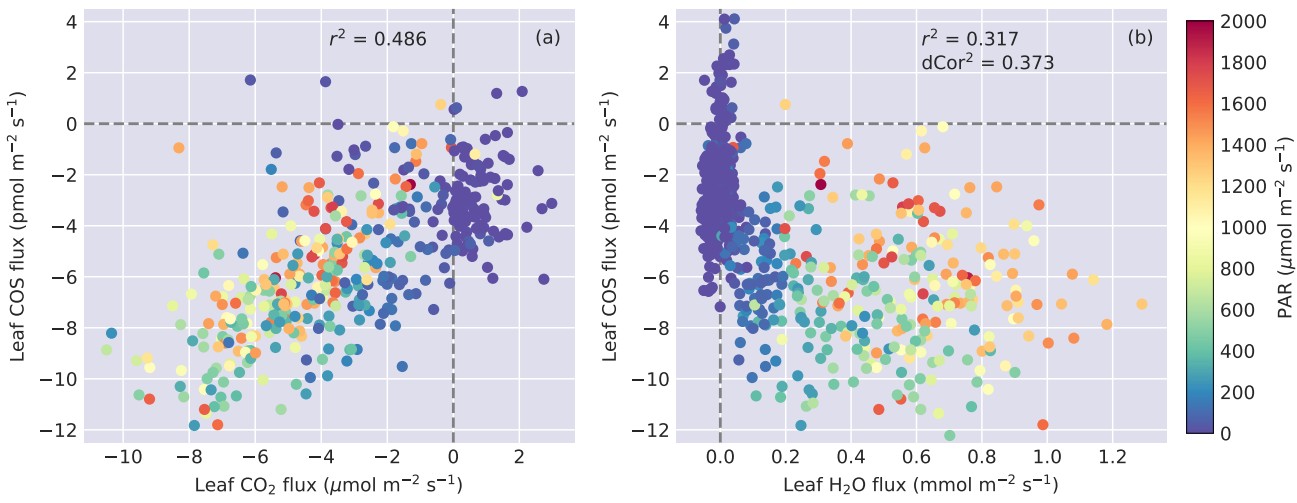

**Figure 4.** (a) Leaf COS vs. $CO_2$ fluxes, and (b) leaf COS vs. $H_2O$ fluxes. Data points are colored by the PAR level.

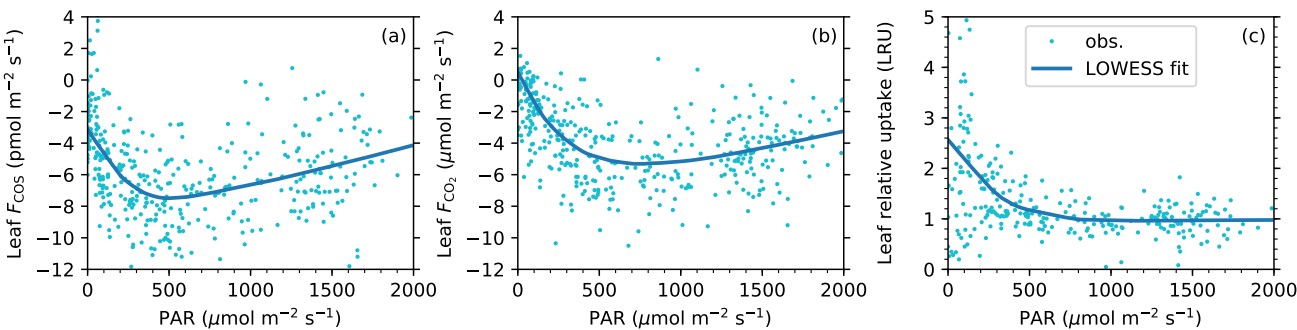

**Figure 5.** Light responses of leaf COS flux (a), $CO_2$ flux (b), and leaf relative uptake ratio (c). Data are shown as dots, and the smoothed curves are fitted with the nonparametric LOWESS method.

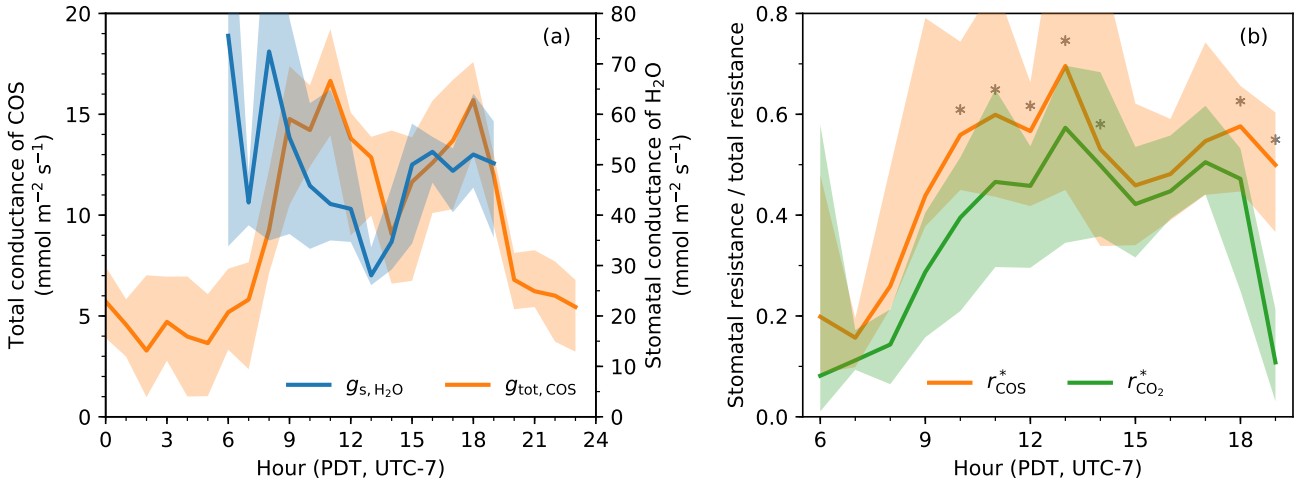

**Figure 6.** (a) Diurnal patterns of the stomatal conductance of water (blue, right $y$-axis) and the total conductance of COS (orange, left $y$-axis). Note that the two variables were on different scales for visual comparison. (b) Daytime patterns of the fraction of stomatal resistance in the total resistance for COS (orange) and for $CO_2$ (green). Similar to Fig. 3, in both panels solid curves indicate medians and shaded areas are between 25th and 75th percentiles, binned by the hour of the day. The asterisk markers in panel (b) indicate that the difference between $r^*_{COS}$ and $r^*_{CO_2}$ for that time of the day is significant at $p < 0.05$ level in a paired two-sample $t$-test.

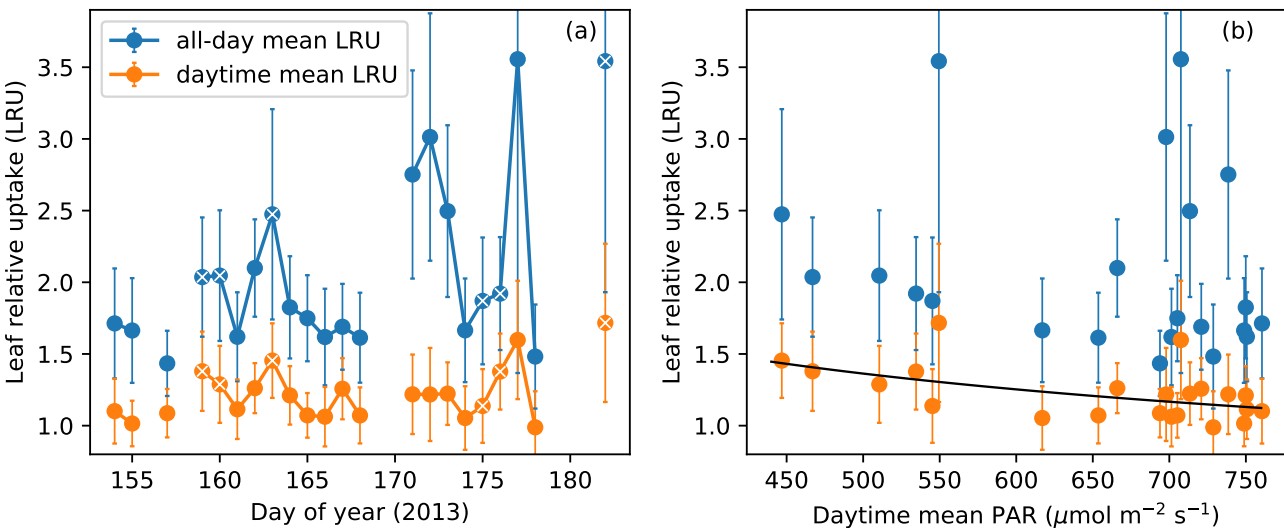

**Figure 7.** (a) All-day mean (blue) and daytime mean (orange) leaf relative uptake (LRU) ratios during the campaign. Data points from overcast days (daytime mean PAR < 550 µmol m$^{-2}$ s$^{-1}$) are labeled with additional white cross signs. (b) All-day mean and daytime mean LRU values vs. daytime mean PAR. Daytime mean LRU vs. PAR follows a response curve (black): LRU = 24.0689 PAR$^{-0.4620}$. Error bars in both panels show ranges of ±1 standard error.

**Table 1.** List of variable symbols

| Symbol | Description |
| --- | --- |
| $\chi_{COS}$ | COS mixing ratio (pptv or pmol mol$^{-1}$) |
| $\chi_{CO_2}$ | CO$_2$ mixing ratio (ppmv or µmol mol$^{-1}$) |
| $\chi_{H_2O}$ | H$_2$O mixing ratio (mmol mol$^{-1}$) |
| $F_{COS}$ | COS flux (pmol m$^{-2}$ s$^{-1}$) |
| $F_{CO_2}$ | CO$_2$ flux (µmol m$^{-2}$ s$^{-1}$) |
| $F_{H_2O}$ | H$_2$O flux (mmol m$^{-2}$ s$^{-1}$) |
| $g_{s,COS}$ | Stomatal conductance of COS (mol m$^{-2}$ s$^{-1}$) |
| $g_{s,CO_2}$ | Stomatal conductance of CO$_2$ (mol m$^{-2}$ s$^{-1}$) |
| $g_{s,H_2O}$ | Stomatal conductance of water (mol m$^{-2}$ s$^{-1}$) |
| $r_{s,COS}$ | Stomatal resistance of COS (mol$^{-1}$ m$^2$ s) |
| $r_{s,CO_2}$ | Stomatal resistance of CO$_2$ (mol$^{-1}$ m$^2$ s) |
| $r_{s,H_2O}$ | Stomatal resistance of water (mol$^{-1}$ m$^2$ s) |
| $g_{tot,COS}$ | Total conductance of COS (mol m$^{-2}$ s$^{-1}$) |
| $g_{tot,CO_2}$ | Total conductance of CO$_2$ (mol m$^{-2}$ s$^{-1}$) |
| $r_{tot,COS}$ | Total resistance of COS (mol$^{-1}$ m$^2$ s) |
| $r_{tot,CO_2}$ | Total resistance of CO$_2$ (mol$^{-1}$ m$^2$ s) |
| $r^*_{CO_2}$ | Ratio of stomatal resistance to total resistance of CO$_2$ |
| $r^*_{COS}$ | Ratio of stomatal resistance to total resistance of COS |
| $T_{ch}$ | Chamber air temperature (°C) |
| $T_{leaf}$ | Leaf temperature (°C) |
| $e_{sat}$ | Saturation vapor pressure (Pa) |
| MFVD or $D$ | Leaf-to-air vapor deficit in mole fraction (mmol mol$^{-1}$) |
| LRU | Instantaneous leaf relative uptake |
| LRU$_{all\text{-}day}$ | All-day mean leaf relative uptake |
| LRU$_{daytime}$ | Daytime mean leaf relative uptake |