# Peer review of "Stomatal control of leaf fluxes of carbonyl sulfide and CO2 in a *Typha* freshwater marsh"

_Biogeosciences, 2017_

## Referee Comment (RC1) · T. E. Gimeno (Referee) · 15 Nov 2017

Sun et al. present here the first field dataset for COS and CO2 leaf relative uptake (LRU) collected in situ during continuous measurement over the peak of a growing season. The authors chose a typical wetland plant (Typha latifolia) and report continuous measurements of CO2 and COS uptake under varying environmental conditions, mainly light (photosynthetically active radiation) and vapour pressure deficit (VPD). They demonstrate that the strong dependency of LRU with PAR observed under laboratory conditions is also observed under natural conditions. The authors explain that strong stomatal control of both processes (COS and CO2 uptake) underlies the observed patterns. Interestingly, the authors report lower LRU values in natural conditions than those previously measured under laboratory conditions. This constitutes a

very valuable contribution as it is the first dataset of LRU collected at the leaf level, in situ, under natural conditions and for more than a month. The paper is very clearly written and the figures and results are nicely presented.

Yet, I do not believe that this paper deserves to be published as regular 'research paper', rather these results would be more appropriately presented as a technical note or rapid report. The reason is that the authors report data from one chamber that measured continuously one single set of six leaves. Their results and conclusions are relevant as they constitute a strong proof of concept, but continuous measurements over a period with limited climatic variability (one campaign with homogenous meteorological conditions, P6L11-15) on a single set of leaves from the same plant (presumably) are not sufficient to constitute a whole research paper. Even more so taking into account the environmental heterogeneity of the light environment (P6L19-21).

In addition I have some major technical concerns and another major concern related to the result interpretation and theoretical framing of the study.

In the methods, the authors claimed that they used a 'flow-through (dynamic) chamber' (P4L5). Yet, during the 5-minute measurement period the chamber acted as a static enclosure (P4L13) and the authors calculated COS and $CO_2$ uptake and transpiration from the slope of the progressive drawdown (or accumulation) of the different species over time (P5L4-15). This approach should be valid provided good mixing, but that is hard to achieve for a >10 L chamber without a fan. The authors need to provide the flow rate entering the cuvette over the measurement period and discuss to what extent they can warranty thorough mixing inside their chamber. In addition, the authors report that they characterised the blank fluxes of (presumably) the same chamber and that these were negligible (methods P4L27). Still, they do not specify how often were these characterised and under what conditions. More important, although it is not stated specifically, it appears that they authors calculated transpiration rates from $H_2O$ vapour concentration measured with the QCLS. If that is the case, I assume the authors did not operate their QCLS coupled to a Nafion drier (or other type of water trap) and

thus they need to correct for the interference associated with the water absorption line (Kooijmans et al. 2016 Atmospheric Measurements Techniques 9:5293-5314).

The authors claim that simultaneous stomatal control of both CO2 and COS uptake underlies the coupling between these processes and the changes in LRU observed under low light and high VPD. Indeed, stomatal control lies at the heart of the discussion and the theoretical framing of the paper, but no data are shown. Also, the authors claimed that they monitored leaf temperature (P4L29-30) and they also had transpiration fluxes (Figure 3c). Still, no calculations of stomatal conductance have been performed. Later in the discussion, some calculations of stomatal conductance are mentioned (P8L20, P9L24), but the authors do not detail how these were obtained. Given that the authors have all the ingredients to calculate stomatal conductance, but yet these are missing, I wonder if this is due to poor mixing inside the chamber, which would have affected all other measurements. This needs to be clarified. In addition, comparing estimates of stomatal conductance derived from COS-uptake measurements with independent quantifications of stomatal conductance from transpiration and leaf temperature would allow to further demonstrate the tight stomatal control of COS uptake. These issues need to be clarified. In addition to these major issues, I have some additional concerns:

Title Remove the term 'Stomatal control' unless you decide to include stomatal conductance measurements, otherwise I suggest "Effects of light and vapour pressure deficit on the coupling of leaf fluxes of carbonyl sulphide and CO2 in a Typha freshwater marsh under natural conditions", or something similar.

Abstract P1L3-4 I think, here, you could be more specific with respect to what we have learned so far: 'LRU is known to increase under low light' P1L15-17 reduce the emphasis on the role of stomatal control.

Introduction This section is interesting and very clearly written. P2L2-10 maybe consider shortening this section, these concepts have already been amply discussed in the literature. P2L2 'COS has been shown to be a unique tracer' P2L8 'The approach

to estimate photosynthesis from COS fluxes' P2L12 'COS and $CO_2$ follow the same diffusional pattern' P2L22 'environmental variables that regulate diffusional limitations, mainly stomatal conductance, including photosynthetically active radiation (PAR) (...) and vapour pressure deficit (VPD)'Also provide a citation here (e.g. Leuning 1995 that you already cite). P2L25 'In contrast to the $CO_2$ flux, at night, COS uptake might continue...' P2L27-28 'Night time COS uptake has been observed.' L29-31 This is not entirely clear. I think here what you mean is that the cited studies inferred vegetation COS uptake from ecosystem-scale measurements instead of direct measurements. Also, please note that both Maseyk et al. (2014) and Commane et al. (2015) found not only evidence for COS uptake, but also emission, this should be briefly mentioned here. P3L16 'We need direct measurements of how LRU...' P3L19-23 Pleas try to specify the research objectives more clearly, or even better formulate two hypotheses (e.g. LRU will decrease under low light in natural conditions) instead of stating the questions that motivated the study.

Methods This section is also very clear and nicely written, but some key details are missing (see major concerns above). P4L5-14 Could you please provide a schematic drawing of the gas-exchange chamber? P4L29-30 Where are the data for leaf temperature? P5L17-18 'Conspicuously unrealistic data points in the meteorological data were removed.' P5L18 'independent criteria to filter measurements' P5L21 'were also discarded' P5L23 'these filtering criteria' P6L2 so if LRU was only calculated during the daytime, why do present the 24-h mean LRU in figure 6?

The results section is very clear and I only have one minor comment: no need to repeat the definition of LRU (P7L1).

Discussion and conclusions In my opinion, this section turned out to be the least interesting of the paper. It is nicely written, but it only consists on a mere repetition of the results and ideas previously presented in the introduction. It can be shortened significantly and I believe the results and discussion section should be merged into one, which would be a much more adequate format for a technical paper. I provide some

further specific details below. P7L19 Provide a citation to support light-independency of COS hydrolysis. P7L17-26 This paragraph is a long compilation of ideas presented already in the results and in the introduction. P7L27-P8L2 this paragraph belongs to the results section. P8L20 provide the details for these calculations in the methods. P9L6-10 Shorten this section, most of these ideas are repeated elsewhere in the paper. P9L16-22 Again repeated ideas, this belongs to the introduction. P9L23 The discussion is not a section appropriate for introducing new equations, move this to the methods. P9L24 detail how this was calculated in the methods. P10L8-10 this discussion on the variation of LRU among species is very relevant. Note that T. latifolia has a very particular physiological behaviour, often exhibiting very high rates of carbon uptake (e.g. Yavitt & Knapp 1998, 139:495-503 or Jespersen et al. 2017 Functional Plant Biology 44:774-784). Thus it is not surprising to find lower LRU value than those previously reported for other plants. Maybe also consider comparing your measurements of leaf CO2 uptake with previous measurements as you seem to have measured much lower values than those previously reported, although this might be simply due to the differences in environmental conditions among studies, most likely light environment. P10L11-24 I do not think it is relevant to discuss the differences between day-time LRU and 24-h averaged LRU. The parameter LRU is useful to estimate GPP from COS uptake, thus it is only relevant during day-time. Please remove this section and the corresponding values from figure 6. P11L2 remove 'that is only stomatal conductance limited'. P11L4-6 rephrase, are 'midday' and 'early afternoon' the same? Because you use them interchangeably here! P11L7-9 I am not quite sure I understand the logic behind this statement. In the afternoon, presumably, PAR does not limit stomatal opening, instead stomatal opening would be limited by high VPD and thus COS and CO2 would both be constraint and hence LRU would not respond to VPD. In fact, I cannot appreciate a change in LRU at midday in figure 3d. I think this conclusion might be a bit misguided by an earlier interpretation of the measurements. P11L10-15 this should be the opening, not the closing paragraph of the discussion.

Figure 4. The data points do not appear colored.

---

## Short Comment (SC1) · 18 Nov 2017

This is a short note to clarify some concerns raised by Reviewer #1. A complete, formal response will follow later.

**Chamber system**

The leaf chamber was not operated as a static enclosure. There was always airflow passing through the chamber. The difference between measurement and non-measurement phase was that the flowrate through the chamber during the measurement period was lower than that provided by the high-speed fan at all other times.

[Figure]

During the measurement period, the flowrate entering the enclosure was the same as that leaving.

In addition to the high-speed ventilation fan, there was a mixing fan in the chamber that was always on to make sure the air inside the chamber was well mixed at all times. See Line 12–14, Page 4 in the Methods section.

We will report more details on the blank chamber measurements as requested in a revised manuscript version. Blank chamber fluxes were negligible.

We did not use a Nafion dryer or other water trap. We applied the water broadening corrections supplied by TDLWintel (see Table 4 and related discussion in Kooijmans et al. 2016).

**Stomatal conductance data**

We do have stomatal conductance estimates, and will add the data, figure, and relevant discussion in the revised manuscript. For now, we have attached a figure of diurnal stomatal conductance estimated from water measurements, and the total conductance of COS (see page C4 of this note).

**Leaf relative uptake (LRU)**

The low LRU observed at this site is not unique to *T. latifolia*. Similar low values were reported by Maseyk et al. (2014) and Commane et al. (2015) for whole-ecosystem relative uptake at high light (1.1 to 1.3, the leaf-level LRU would likely be even lower). We have also seen unpublished data from other sites showing similar low LRU values under high light. Therefore we don't think the low LRU is related to something unusual

about this species.

We have included data on calculated 24-hour mean LRU as it is relevant for large-scale applications of COS, e.g., the inverse estimation of terrestrial gross $CO_2$ uptake from airborne COS measurements. In large-scale applications, daytime and nighttime air masses are mixed and cannot be separated. Since nighttime COS uptake can happen, which leads to different patterns of boundary layer COS and $CO_2$ concentrations compared with the daytime, it must be considered in the surface COS budget in simulations of atmospheric transport models. The 24-hour mean LRU would then be a diagnostic of simulated COS versus $CO_2$ drawdown patterns. We will aim to clarify the reason for including the 24-hour LRU in the revised version.

―――――――――――――――――――

[Figure]

**Fig. 1.** Hourly median stomatal conductance of water (blue; right axis) and total conductance of COS (orange; left axis). The shaded areas indicate the ranges between 25% and 75% percentiles.

---

## Referee Comment (RC2) · M.E. Whelan (Referee) · 3 Jan 2018

Leaf relative uptake (LRU) of COS and CO2 is a parameter that is often used to estimate plant CO2 uptake from observed ecosystem fluxes of COS. There are other sources and sinks of COS in ecosystems, though they are typically small compared to uptake through plant stomata. One important exception is wetland soils, which tend to be a relatively large source of COS. In non-wetland or agricultural systems, measurements of net CO2 and COS concentrations and fluxes are sufficient to make an estimate of GPP with an approximation of LRU.

Here, Sun et al. present a dataset of H2O, COS, and CO2 flux and concentration measurements from a single leaf chamber in a wetland over about 36 days. This type of

data is an important contribution and will be undoubtedly useful for other studies. However, the interpretation would be aided by greater attention to stomatal conductance, as the title implies, rather than LRU.

The trouble with focusing on LRU is the matter of scale and applicability. Work by Hilton et al. (2015) demonstrated that, for regional GPP estimates, LRU is not the most important source of uncertainty. On the leaf scale, a direct measurement of $CO_2$ uptake can be made, though it includes photorespiration. At the tower-level scale (1 km2), I am not sure that COS-based GPP estimates are more accurate than recent approaches relying on $CO_2$ measurements alone, though the Wehr et al. (2017) study in a temperate forest demonstrated COS-based estimates of canopy stomatal conductance were consistent with other measurement approaches in that system. In short, LRU is not the most important question on large scales, not employed in and of itself on leaf scales, and has some applicability still under development at the site scale. While having a better description of LRU variation with PAR would be an improvement, it is not the urgent next step that the text here describes.

The second issue is applicability to other ecosystems. This dataset was collected from a chamber containing leaves of a plant typically found in wetlands. The COS-GPP tracer technique is not usually applied at the site level in wetlands because of often substantial COS production from wetland soils. Also, some wetland plants have interesting adaptations to tolerate suboxic soil environments. For example, Typha have well developed aerenchyma to allow oxygen to diffuse into the root zone. Aerenchyma can also transport reduced gas compounds to the surface, circumventing oxidation in the water column. This has been shown for methane and Whelan et al., (2013) suggested a similar route for carbonyl sulfide. The data do not necessarily show COS release from the parts of the leaves enclosed in the chamber, bu teasing apart uptake from other sources of COS in the system would probably be a challenge. It is confusing to carry out an LRU study in one of the few ecosystems where applying LRU to back out GPP is an exception to the simplicity of the approach.

**BGD**

While using LRU is probably the most popular method of calculating GPP from COS measurements, it is not the only method. The SiB model, for instance, has a "mechanistic" uptake representation that does not rely on an LRU number. The applicability of COS measurements to carbon cycle studies does not depend solely on LRU.

Motivating this study interpretation with the vagaries of leaf conductances would be of greater interest. Already, Sun et al. show that nighttime stomatal conductance is occurring and that daytime conductances change with evaporative demand. Sect. 4.2 should be expanded to include the broader literature on nocturnal stomatal conductance, rather than restricting the discussion to focus only on COS studies. Graphically comparing an established method to the COS-based method of estimating stomatal conductance could reveal possible mismatches and highlight the strengths of each approach, even if leaf temperature was not measured precisely. Re-working the figures to this effect would be beneficial.

Small technical concerns include publishing chamber blank results and also the exact equation that was used for the QCL water correction. There are a growing number of researchers using this make of QCL and water is a problem for the older models.

In short, this is a good dataset, but the interpretation could perhaps avoid the concept of LRU entirely.

Sincerely,

Mary Whelan

---

## Author Response (AR1)

Dear Editor,

We have addressed all concerns raised in the comments of both reviewers in the revision. Per your request, below is a concise list of relevant changes made in the manuscript. For the detailed point-by-point response, please see Page 3. We appreciate your consideration of the revised manuscript.

**List of changes in the revised manuscript**

"R1" and "R2" refer to concerns raised by Reviewers #1 (T.E.G) and #2 (M.E.W.), respectively. Line and page numbers refer to locations in the revised manuscript, not the marked-up document for change tracking.

1. Abstract
   - The second paragraph has been revised to give an accurate account of the role of stomatal control (R1).
   - Language correction has been made in P1L4 (R1).
2. Introduction
   - The first paragraph has been shortened (R1). See P2L2–L8.
   - Language corrections have been made (R1).
   - The research objectives have been redefined in terms of hypotheses in P3L14–L20 (R1).
3. Methods
   - A schematic figure of the chamber (Fig. 1a) has been provided (R1).
   - We have clarified that the chamber was not operated as a static enclosure during the measurement period (R1). See also Fig. 1b.
   - Details on how the fans inside the chamber were operated have been added in P4L8–L13 (R1).
   - The flow rate and the chamber turnover time have been added in P4L17–L18 (R1).
   - We have added blank chamber effects in P4L32–L34, with more details provided also in §S2 of the Supplement (R1 & R2).
   - A description of the water correction of QCL measurements has been added in P4L23–L27 (R1 & R2). More details have been supplied in §S1 of the Supplement.
   - Leaf temperature data have been shown in Fig. S3 in the Supplement and added to the online dataset (R1).
   - §2.5 has been added to describe the calculations of various flux-derived variables, including the stomatal conductance and time-integrated LRUs (R1 & R2).

- Language corrections have been made (R1).

4. Results
   - Stomatal conductance estimates have been added and described in details (R1 & R2). See Fig. 6 and §3.2.
   - A distinction between the instantaneous LRU and time-integrated LRUs has been made in §2.5.2 and §3.3 (R1).

5. Discussion
   - Redundant texts in this section have all been removed (R1). Paragraphs that belong to the Results section have been relocated there (R1).
   - The former §4.1 has been removed (R1).
   - The main idea in former §4.3 has now been addressed in the new §4.1, which has been completely rewritten to give a proper mechanistic discussion of the environmental controls on LRU (R1 & R2).
   - An explanation of why high VPD drives LRU lower has been given in P10L25–L29 in §4.1 (R1).
   - §4.2 and §4.3 have been condensed significantly (R1).
   - The calculation of nighttime stomatal conductance has been documented in §S4 of the Supplement (R1).
   - We have given a mechanistic explanation of the low LRU observed on *T. latifolia* in P11L33–P12L3 in §4.3 (R1).
   - The rationale behind the use of all-day mean LRU has been discussed in P12L7–L15 (R1).
   - The confusion about 'midday' and 'early afternoon' has been eliminated (R1).
   - The limitation of the LRU method in large-scale GPP estimation has been discussed in P12L7–L15 in §4.3 (R2).

6. The Conclusion has been completely rewritten.

7. Figures
   - A schematic figure of the chamber is added to Fig. 1a.
   - Fig. 3: The $y$-axis range in panel d has been changed to emphasize LRU variation; in panel f, vapor deficit has been corrected with respect to leaf temperature.
   - Fig. 4 has been revamped for better visualization of the colored data points.
   - Fig. 6 has been added to show stomatal conductance data.

8. Table 1: A list of variable symbols has been added.

9. The Supplement includes additional information on the water correction of QCL measurements (§S1), blank chamber effects (§S2), leaf temperature data (§S3), and estimation of the nighttime stomatal conductance (§S4).

**Response to the comments on "Stomatal control of leaf fluxes of carbonyl sulfide and $CO_2$ in a *Typha* freshwater marsh"**

Wu Sun, on behalf of all coauthors

In this response, the text is formatted as: referees' comments (indented blocks), authors' response, **applied changes in the manuscript**. Locations in the revised manuscript are referenced by "P{#X}L{#Y}", meaning "Line Y on Page X".

**Reply to comments by Teresa E. Gimeno (Referee #1)**

> Sun et al. present here the first field dataset for COS and $CO_2$ leaf relative uptake (LRU) collected in situ during continuous measurement over the peak of a growing season. The authors chose a typical wetland plant (*Typha latifolia*) and report continuous measurements of $CO_2$ and COS uptake under varying environmental conditions, mainly light (photosynthetically active radiation) and vapour pressure deficit (VPD). They demonstrate that the strong dependency of LRU with PAR observed under laboratory conditions is also observed under natural conditions. The authors explain that strong stomatal control of both processes (COS and $CO_2$ uptake) underlies the observed patterns. Interestingly, the authors report lower LRU values in natural conditions than those previously measured under laboratory conditions. This constitutes a very valuable contribution as it is the first dataset of LRU collected at the leaf level, in situ, under natural conditions and for more than a month. The paper is very clearly written and the figures and results are nicely presented.

We thank Dr. Gimeno for her evaluation of our manuscript. We have made corrections and clarifications to concerns raised in the comments. Below please find a detailed point-by-point response.

> Yet, I do not believe that this paper deserves to be published as regular 'research paper', rather these results would be more appropriately presented as a technical note or rapid report. The reason is that the authors report data from one chamber that measured continuously one single set of six leaves. Their results and conclusions are relevant as they constitute a strong proof of concept, but continuous measurements

over a period with limited climatic variability (one campaign with homogenous me-
teorological conditions, P6L11–15) on a single set of leaves from the same plant (pre-
sumably) are not sufficient to constitute a whole research paper. Even more so taking
into account the environmental heterogeneity of the light environment (P6L19–21).

We acknowledge these limitations in data collection; however, none of them should weaken the
main conclusions or disqualify the manuscript from a full research paper. In fact, the experiments
were designed to characterize leaf relative uptake (LRU) variability in response to environmental
controls, particularly on the diurnal timescale. It represented the first continuous measurement
of leaf fluxes under field conditions, and therefore the first validation of theory and lab measure-
ments. This experimental design in turn served the higher purpose of COS-based ecosystem GPP
estimation by providing accurate LRU parameters.

An ideal experimental design entails randomization and replication, yet field conditions and the
resources available can often be restrictive. We were limited by the sampling time on the QCLS
analyzer, because in each hour, 45 minutes were allotted to eddy covariance measurements (un-
published) and the rest was divided between a soil chamber (unpublished) and a leaf chamber
(reported here). To support COS-based GPP estimation *on the hourly timescale*, LRU measurements
must have the same resolution in time. This was the main reason that a high sampling frequency
was chosen at the expense of having multiple leaf chambers. Nevertheless, we had a large sample
size ($N > 300$) to support a robust analysis of LRU variability.

The similarity in day-to-day meteorological conditions was a blessing rather than a defect of the
study, because it means that, other than PAR and vapor deficit, little else was changing, creating
an ideal situation for testing LRU responses to PAR and vapor deficit. Indeed, diurnal variations of
fluxes and LRU in response to PAR and vapor deficit were well characterized with a high sampling
frequency. Besides, there were a few overcast days that caused the daytime mean LRU to increase;
this was made clear in Fig. 7. The PAR sensor used in the study was collocated with the chamber
(see P5L3–L4 added in the revised text), and this should have properly accounted for the light
microenvironment around the chamber.

In addition I have some major technical concerns and another major concern related
to the result interpretation and theoretical framing of the study.

**We have made revisions and added the requested information in the Methods section to**
**address these technical concerns. We have provided stomatal conductance estimates to**

**improve data interpretations and to better support the main messages of this study.**

In the methods, the authors claimed that they used a 'flow-through (dynamic) chamber' (P4L5). Yet, during the 5-minute measurement period the chamber acted as a static enclosure (P4L13) and the authors calculated COS and $CO_2$ uptake and transpiration from the slope of the progressive drawdown (or accumulation) of the different species over time (P5L4–15). This approach should be valid provided good mixing, but that is hard to achieve for a > 10 L chamber without a fan. The authors need to provide the flow rate entering the cuvette over the measurement period and discuss to what extent they can warranty thorough mixing inside their chamber. In addition, the authors report that they characterised the blank fluxes of (presumably) the same chamber and that these were negligible (methods P4L27). Still, they do not specify how often were these characterised and under what conditions. More important, although it is not stated specifically, it appears that they authors calculated transpiration rates from $H_2O$ vapour concentration measured with the QCLS. If that is the case, I assume the authors did not operate their QCLS coupled to a Nafion drier (or other type of water trap) and thus they need to correct for the interference associated with the water absorption line (Kooijmans et al. 2016 Atmospheric Measurements Techniques 9: 5293–5314).

These technical concerns have been addressed in §2.2 Experimental setup.

The leaf chamber was not operated as a static enclosure. There was always airflow passing through the chamber, supplied by a vacuum pump. During the measurement phase, the flow rate entering the enclosure was the same as that leaving. To clarify that the chamber was always an open system, **the label of the measurement phase has been changed from "ch closed" to "ch meas" in Fig. 1b**.

There were two fans running in the chamber, one for ventilation, and the other for mixing. During the measurement phase indicated in Fig. 1b, the ventilation fan was turned off, but the chamber nevertheless was still a flow-through system, because (i) the pump was pulling air from the chamber and (ii) the opening of the ventilation fan served as the inlet. The mixing fan was kept running continuously to make sure the air inside the chamber was always mixed. **This paragraph has been rewritten.** See P4L8–L13 in §2.2 Experimental setup.

The median flow rate through the chamber was 6.4 slm, which translated to a chamber turnover

time of 1.5 minutes. **This information has been added in P4L17–L18.**

Blank chamber effects were negligible: $0.05 \pm 0.29$ pmol m$^{-2}$ s$^{-1}$ for COS and $0.02 \pm 0.15$ μmol m$^{-2}$ s$^{-1}$ for $CO_2$. **This information has been added in P4L32–L34.** Please also see the Supplement for a detailed description.

We did not use a Nafion dryer or any other water trap. We applied the water broadening corrections supplied by TDLWintel—a data acquisition software on the QCL—using default correction factors. **We have now described the water correction procedure in P4L23–L27.** In addition, we have also discussed the potential influences of the uncertainty in the $CO_2$ water correction factor (see the Supplement).

> The authors claim that simultaneous stomatal control of both $CO_2$ and COS uptake underlies the coupling between these processes and the changes in LRU observed under low light and high VPD. Indeed, stomatal control lies at the heart of the discussion and the theoretical framing of the paper, but no data are shown. Also, the authors claimed that they monitored leaf temperature (P4L29–30) and they also had transpiration fluxes (Figure 3c). Still, no calculations of stomatal conductance have been performed. Later in the discussion, some calculations of stomatal conductance are mentioned (P8L20, P9L24), but the authors do not detail how these were obtained. Given that the authors have all the ingredients to calculate stomatal conductance, but yet these are missing, I wonder if this is due to poor mixing inside the chamber, which would have affected all other measurements. This needs to be clarified. In addition, comparing estimates of stomatal conductance derived from COS-uptake measurements with independent quantifications of stomatal conductance from transpiration and leaf temperature would allow to further demonstrate the tight stomatal control of COS uptake. These issues need to be clarified.

**Diurnal patterns of the stomatal conductance of $H_2O$ and the total conductance of COS have now been presented in the new Figure 6. Interpretations and discussions of the results have been added.** See §3.2 in the Results and §4.1 in the Discussions. We have also described how stomatal conductance of $H_2O$ and the total conductance of COS are calculated in §2.5.1 in the Methods.

> In addition to these major issues, I have some additional concerns:

Title Remove the term 'Stomatal control' unless you decide to include stomatal conductance measurements, otherwise I suggest "Effects of light and vapour pressure deficit on the coupling of leaf fluxes of carbonyl sulphide and $CO_2$ in a *Typha* freshwater marsh under natural conditions", or something similar.

**We have included stomatal conductance estimates. The title is kept unchanged.**

Abstract

P1L3–4 I think, here, you could be more specific with respect to what we have learned so far: 'LRU is known to increase under low light'.

**Revised.** See P1L4.

P1L15–17 reduce the emphasis on the role of stomatal control.

Since stomatal conductance data have been added to the revised manuscript, the emphasis on the role of stomatal control is appropriate.

Introduction

This section is interesting and very clearly written.

P2L2–10 maybe consider shortening this section, these concepts have already been amply discussed in the literature.

**We have shortened this paragraph by 25%.** See P2L2–L8.

P2L2 'COS has been shown to be a unique tracer'.

**Changed to "Carbonyl sulfide (COS) is a unique tracer for . . .".** See P2L2.

P2L8 'The approach to estimate photosynthesis from COS fluxes'

**This sentence has been removed for conciseness.**

P2L12 'COS and $CO_2$ follow the same diffusional pattern'

**We think that 'pathway' is a more suitable word.** A search in Google Ngram (`https://books.google.com/ngrams/`) finds no result of the phrase 'diffusional pattern'.

P2L22 'environmental variables that regulate diffusional limitations, mainly stomatal conductance, including photosynthetically active radiation (PAR) (. . .) and vapour pressure deficit (VPD)' Also provide a citation here (e.g. Leuning 1995 that you already cite).

**Revised according to the reviewer's suggestion.** See P2L18–L21.

P2L25 'In contrast to the $CO_2$ flux, at night, COS uptake might continue. . .'

**Revised to "At night, in contrast to the $CO_2$ emission, COS uptake may continue . . .".** See P2L22.

P2L27–28 'Night time COS uptake has been observed..'

**Revised: 'found' —> 'observed'.** See P2L24.

L29–31 This is not entirely clear. I think here what you mean is that the cited studies inferred vegetation COS uptake from ecosystem-scale measurements instead of direct measurements.

**The sentence has been improved to clarify the point.** See P2L26–L27.

Also, please note that both Maseyk et al. (2014) and Commane et al. (2015) found not only evidence for COS uptake, but also emission, this should be briefly mentioned here.

Emissions reported in Maseyk et al. (2014) came from soils and mature grain heads. And those in Commane et al. (2015) were not found in subsequent years of their campaign (Wehr et al., 2017). Since the scope of this study is leaf scale COS exchange, non-foliar sources of COS are only of peripheral importance and are hence not elaborated here.

P3L16 'We need direct measurements of how LRU. . .'

**Revised.** See P3L12.

P3L19–23 Please try to specify the research objectives more clearly, or even better formulate two hypotheses (e.g. LRU will decrease under low light in natural conditions) instead of stating the questions that motivated the study.

**The research objectives have been rephrased in terms of clearly defined hypotheses.** See P3L14–L20.

Methods

This section is also very clear and nicely written, but some key details are missing (see major concerns above).

The missing details have been added.

P4L5–14 Could you please provide a schematic drawing of the gas-exchange chamber?

A schematic diagram of the chamber has been added in panel (a) of the new Figure 1.

P4L29–30 Where are the data for leaf temperature?

**Leaf temperature data are now shown in Figure S3 in the Supplement and are added to the online dataset. In addition, vapor deficit shown in Fig. 3f has been corrected with respect to leaf temperature.**

P5L17–18 'Conspicuously unrealistic data points in the meteorological data were removed.'
P5L18 'independent criteria to filter measurements'
P5L21 'were also discarded'
P5L23 'these filtering criteria'

**Revised following the referee's suggestions.** See P5L21, L22, L25, and L27.

Results

P6L2 so if LRU was only calculated during the daytime, why do present the 24-h mean LRU in figure 6?

**We have now clarified how the instantaneous LRU (i.e., the commonly referred 'LRU' in the literature) and the time-integrated LRU are calculated in §2.5.2 in the Methods.** The use of time-integrated LRU is relevant to large-scale applications, as is discussed in §4.3 in the Discussions.

The results section is very clear and I only have one minor comment: no need to repeat the definition of LRU (P7L1).

**We have removed the redundant definition.**

Discussion and conclusions

In my opinion, this section turned out to be the least interesting of the paper. It is nicely written, but it only consists on a mere repetition of the results and ideas previously presented in the introduction. It can be shortened significantly and I believe the results and discussion section should be merged into one, which would be a much more adequate format for a technical paper. I provide some further specific details below.

**We have restructured and greatly abridged the discussion to strive for a balance between conciseness and clarity. The former §4.1 has been removed, because most of its original contents are now addressed in the Results. The former §4.3 on LRU environmental control has been completely rewritten, explained in terms of the stomatal vs. internal conductance competition. Other parts of the discussion have been condensed significantly.**

For the optimal flow of the text, we did not merge the Discussion into the Results. This is simply because each part of the Discussion may rely on multiple pieces of information from the Results.

P7L19 Provide a citation to support light-independency of COS hydrolysis.

**Added Protoschill-Krebs et al. (1996).** See P10L15.

P7L17–26 This paragraph is a long compilation of ideas presented already in the results and in the introduction.

**Removed.**

P7L27–P8L2 this paragraph belongs to the results section.

**Moved to §3.1 in the Results.** See P8L28–L33.

P8L20 provide the details for these calculations in the methods.

This is intended as a back-of-the-envelope calculation for discussion only. Strictly speaking, the obtained value is an estimate derived from the data, so it is not appropriate to document the calculations in the methods. **We have detailed the calculations in the Supplement instead.**

P9L6–10 Shorten this section, most of these ideas are repeated elsewhere in the paper.

**Removed.**

P9L16–22 Again repeated ideas, this belongs to the introduction.

**This part has been completely rewritten.** See P10L20–L30 in §4.1.

P9L23 The discussion is not a section appropriate for introducing new equations, move this to the methods.

P9L24 detail how this was calculated in the methods.

**This part has been removed** since it is no longer essential for the discussion. However, generally, there is no rule or guideline to discourage the use of equations in the discussion, if they are well explained.

P10L8–10 this discussion on the variation of LRU among species is very relevant. Note that *T. latifolia* has a very particular physiological behaviour, often exhibiting very

high rates of carbon uptake (e.g. Yavitt & Knapp 1998, 139:495–503 or Jespersen et al. 2017 Functional Plant Biology 44:774–784). Thus it is not surprising to find lower LRU value than those previously reported for other plants. Maybe also consider comparing your measurements of leaf $CO_2$ uptake with previous measurements as you seem to have measured much lower values than those previously reported, although this might be simply due to the differences in environmental conditions among studies, most likely light environment.

The reviewer raised an interesting point regarding the link between LRU and photosynthetic parameters. **We have added a brief explanation to the low LRU of the *T. latifolia* in P11L33–P12L3 in §4.3.**

P10L11–24 I do not think it is relevant to discuss the differences between day-time LRU and 24-h averaged LRU. The parameter LRU is useful to estimate GPP from COS uptake, thus it is only relevant during day-time. Please remove this section and the corresponding values from figure 6.

The all-day mean LRU is relevant to large-scale applications, because regional COS drawdown patterns are time-integrated features. **This is discussed in P12L7–L15 in §4.3.**

P11L2 remove 'that is only stomatal conductance limited'.

**Removed.**

P11L4–6 rephrase, are 'midday' and 'early afternoon' the same? Because you use them interchangeably here!

**This sentence has been removed from the conclusion. We have taken care to use these terms consistently in other parts.**

P11L7–9 I am not quite sure I understand the logic behind this statement. In the afternoon, presumably, PAR does not limit stomatal opening, instead stomatal opening would be limited by high VPD and thus COS and $CO_2$ would both be constraint and hence LRU would not respond to VPD.

This is because COS uptake is more stomatal-conductance-limited than $CO_2$ uptake due to the

much higher enzyme activity of CA in catalyzing COS hydrolysis ($k_{cat}/K_m$ of CA $> k_{cat}/K_m$ of RuBisCO). **We have added a discussion of this issue in P10L25–L29 in §4.1.**

In fact, I cannot appreciate a change in LRU at midday in figure 3d. I think this conclusion might be a bit misguided by an earlier interpretation of the measurements.

**The $y$-axis range of Figure 3d has been adjusted to emphasize the variations of LRU.** A "dip" of LRU between 15:00 and 18:00 should be clearly visible now.

P11L10–15 this should be the opening, not the closing paragraph of the discussion.

This paragraph no longer exists. **The conclusion has been rewritten.**

Figure 4. The data points do not appear colored.

**The figure has been revamped to resolve the issue experienced by the reviewer.** The problem was likely caused by the aliasing of the edges of data points under low-resolution conditions.

**Reply to comments by Mary E. Whelan (Referee #2)**

Leaf relative uptake (LRU) of COS and $CO_2$ is a parameter that is often used to estimate plant $CO_2$ uptake from observed ecosystem fluxes of COS. There are other sources and sinks of COS in ecosystems, though they are typically small compared to uptake through plant stomata. One important exception is wetland soils, which tend to be a relatively large source of COS. In non-wetland or agricultural systems, measurements of net $CO_2$ and COS concentrations and fluxes are sufficient to make an estimate of GPP with an approximation of LRU.

We thank Dr. Whelan for her helpful and insightful comments. Indeed, COS fluxes from wetland soils—potentially large sources—need to be carefully constrained when using COS measurements to infer GPP. In this study, the chamber enclosure created a separate system for leaf gas exchange that was free from soil interference. When scaled up to the canopy, with soil COS budget constrained, the COS method for GPP estimation can still work reliably in a wetland ecosystem. The

treatment of soil COS budget in GPP estimation is out of the scope of this paper, but will be demonstrated in a manuscript on ecosystem-scale COS fluxes by our group (Seibt et al., in prep.).

> Here, Sun et al. present a dataset of $H_2O$, COS, and $CO_2$ flux and concentration measurements from a single leaf chamber in a wetland over about 36 days. This type of data is an important contribution and will be undoubtedly useful for other studies. However, the interpretation would be aided by greater attention to stomatal conductance, as the title implies, rather than LRU.

**We have provided data and a figure of stomatal conductance estimates at the request of both reviewers.** See Figure 6 and §2.5.1 in the revised manuscript.

We acknowledge that the previous version might have created misleading expectations for the study of LRU. **We have rewritten most of the Results and the Discussions to reorient the manuscript on how LRU varies in field conditions and how such behavior manifests stomatal responses.**

> The trouble with focusing on LRU is the matter of scale and applicability. Work by Hilton et al. (2015) demonstrated that, for regional GPP estimates, LRU is not the most important source of uncertainty. On the leaf scale, a direct measurement of $CO_2$ uptake can be made, though it includes photorespiration. At the tower-level scale (1 km$^2$), I am not sure that COS-based GPP estimates are more accurate than recent approaches relying on $CO_2$ measurements alone, though the Wehr et al. (2017) study in a temperate forest demonstrated COS-based estimates of canopy stomatal conductance were consistent with other measurement approaches in that system. In short, LRU is not the most important question on large scales, not employed in and of itself on leaf scales, and has some applicability still under development at the site scale. While having a better description of LRU variation with PAR would be an improvement, it is not the urgent next step that the text here describes.

At the ecosystem scale ($\sim$1 km$^2$), the COS-based method for GPP estimation is meant to supplement rather than replace conventional $CO_2$-based methods. In terms of accuracy, it is true that previous studies that applied the COS method on the ecosystem scale ended up getting similar—but not more accurate—results compared with the $CO_2$-based methods. Part of the reason was that LRU variability was unable to be treated properly due to the lack of concurrent leaf-level measurements. The uncertainty in LRU would further propagate into the GPP estimates. Recognizing the problem, our study aims to contribute to its solution rather than circumvent it.

Yet the actual value-added benefit of COS tracer lies in the fact that it provides GPP estimates that are *independent* of assumptions on the temperature response of respiration and on the light response of photosynthesis—at least one of which is required in $CO_2$-based methods (Reichstein et al., 2005; Lasslop et al., 2010). In other words, uncertainties in the built-in assumptions of $CO_2$-based methods cannot be assessed unless other *independent constraints*—such as COS—are introduced. For example, COS-based GPP estimates may allow us to obtain daytime respiration straightforwardly, which further opens the possibility of studying the Kok effect (i.e., light inhibition of leaf respiration). Various ecophysiological applications of COS form an evolving frontier, and the usefulness of COS could not be overstated.

At large scales, currently available datasets are limited in spatial and temporal coverage, and the uncertainty in remotely retrieved COS concentrations would likely overwhelm the uncertainty of LRU in GPP-oriented applications (Whelan et al., 2017). But the research field likely would not stay there. Were better COS data products to be available in the future to allow for data assimilation at finer spatial and temporal scales—like a 'NOAA CarbonTracker' for COS—then LRU would become an issue. LRU responses to light and VPD would mean that synoptic weather events may shift regional estimates of daily averaged LRU. Without accounting for the relevant effects on LRU, GPP products derived from COS measurements could be biased. Although models like SiB can simulate LRU ab initio, the simulated LRU has yet to be validated with field studies and it is too early to put complete trust in model-generated LRU values.

In short, the outstanding issues around LRU have been underappreciated; but it does not mean that LRU is well understood and it no longer begs for questions, nor would LRU be less useful with the presence of process-based models. LRU is still an indispensable tool linking COS and $CO_2$ uptake, because it is simple enough to provide an understanding of the relationship between COS and $CO_2$ uptake, which would otherwise be inscrutable.

> The second issue is applicability to other ecosystems. This dataset was collected from a chamber containing leaves of a plant typically found in wetlands. The COS–GPP tracer technique is not usually applied at the site level in wetlands because of often substantial COS production from wetland soils. Also, some wetland plants have interesting adaptations to tolerate suboxic soil environments. For example, *Typha* have well developed aerenchyma to allow oxygen to diffuse into the root zone. Aerenchyma can also transport reduced gas compounds to the surface, circumventing oxidation

in the water column. This has been shown for methane and Whelan et al., (2013) suggested a similar route for carbonyl sulfide. The data do not necessarily show COS release from the parts of the leaves enclosed in the chamber, but teasing apart uptake from other sources of COS in the system would probably be a challenge. It is confusing to carry out an LRU study in one of the few ecosystems where applying LRU to back out GPP is an exception to the simplicity of the approach.

During the same campaign, we had a soil chamber installed to characterize soil COS emissions. We have already attempted COS-based GPP estimation at the site. The COS-based GPP estimates ($GPP_{COS}$) agree well with traditional $CO_2$-based GPP estimates ($GPP_{NEE}$). Results from that study have been presented at the 2016 AGU Fall Meeting, and are currently being written up as a manuscript (Seibt et al., in prep.).

As for the aerenchymal COS transport, we did not have the means to measure its contribution to COS fluxes. However, the close resemblance between $GPP_{COS}$ and $GPP_{NEE}$ suggests that the aerenchymal COS transport does not constitute a significant missing source of COS, although its presence cannot be ruled out.

While using LRU is probably the most popular method of calculating GPP from COS measurements, it is not the only method. The SiB model, for instance, has a "mechanistic" uptake representation that does not rely on an LRU number. The applicability of COS measurements to carbon cycle studies does not depend solely on LRU.

This is a valid point. But the advantages of SiB shine better in large-scale applications, especially when a representative LRU is difficult to determine from the upscaling of field data. **We have revised the related discussion in the manuscript to reflect this point (P12L7–L15).**

Motivating this study interpretation with the vagaries of leaf conductances would be of greater interest. Already, Sun et al. show that nighttime stomatal conductance is occurring and that daytime conductances change with evaporative demand. Sect. 4.2 should be expanded to include the broader literature on nocturnal stomatal conductance, rather than restricting the discussion to focus only on COS studies. Graphically comparing an established method to the COS-based method of estimating stomatal conductance could reveal possible mismatches and highlight the strengths of each approach, even if leaf temperature was not measured precisely. Re-working the figures to this effect would be beneficial.

We thank Dr. Whelan's suggestions for improvement. The following changes have been made to address these issues:

- **Figure 6 (new) has been added to show diurnal trends of the stomatal conductance of water and the total conductance of COS.**
- **In §4.2, the nighttime stomatal conductance estimate has been corrected for an erroneous assumption that internal conductance is negligible. The discussion has also been improved.**

Small technical concerns include publishing chamber blank results and also the exact equation that was used for the QCL water correction. There are a growing number of researchers using this make of QCL and water is a problem for the older models.

**Blank chamber effects are now provided in §2.2 Experimental setup.** See P4L32–L34. Please also see the Supplement for more details on the blank chamber effects.

**We have added information on the QCL water correction.** See P4L23–L27 in §2.2. A detailed description of the equations used for water vapor correction and their effects on flux uncertainty is given in the Supplement.

In short, this is a good dataset, but the interpretation could perhaps avoid the concept of LRU entirely.

We have explained why LRU is useful in assessing the relationship between COS and $CO_2$ leaf uptake. Please see the reply to a previous comment on Pages 14–15 of this response.

**Stomatal control of leaf fluxes of carbonyl sulfide and CO$_2$ in a *Typha* freshwater marsh**

Wu Sun[1], Kadmiel Maseyk[2,a], Céline Lett[3,a], and Ulli Seibt[1,a]

[1]Department of Atmospheric and Oceanic Sciences, University of California, Los Angeles, CA 90095-1565, USA
[2]School of Environment, Earth and Ecosystem Sciences, The Open University, Milton Keynes MK7 6AA, United Kingdom
[3]Laboratoire des Sciences du Climat et de l'Environnement, Université Paris Saclay, 91191 Gif-sur-Yvette, France
[a]formerly at Institute of Ecology and Environmental Sciences, Université Pierre et Marie Curie Paris 6, France

**Correspondence:** Wu Sun (wu.sun@ucla.edu) and Ulli Seibt (useibt@ucla.edu)

**Abstract.** Carbonyl sulfide (COS) is an emerging tracer to constrain land photosynthesis at canopy to global scales, because leaf COS and CO$_2$ uptake processes are linked through stomatal diffusion. The COS tracer approach requires knowledge of the concentration normalized ratio of COS uptake to photosynthesis, commonly known as the leaf relative uptake (LRU). LRU is known to [..[1] ]increase under low light, but the environmental controls over LRU variability in the field are poorly understood
5   due to scant leaf scale observations.

Here we present the first direct observations of [..[2] ]LRU responses to environmental variables in the field. We measured leaf COS and CO$_2$ fluxes at a freshwater marsh in summer 2013. Daytime leaf COS and CO$_2$ uptake showed similar peaks in the mid-morning and late afternoon [..[3] ]separated by a prolonged midday depression, highlighting the common stomatal control on [..[4] ]diffusion. At night, in contrast to CO$_2$, COS uptake continued, indicating partially open stomata. LRU ratios
10   showed a clear relationship with photosynthetically active radiation (PAR), converging to 1.0 at high PAR, while increasing sharply at low PAR. Daytime integrated LRU (calculated from daytime mean COS and CO$_2$ uptake) ranged from 1 to 1.5, with a mean of 1.2 across the campaign, significantly lower than [..[5] ]previously reported laboratory mean value (~1.6). Our results indicate two major determinants of LRU—light and vapor [..[6] ]deficit. Light is the primary driver of LRU because CO$_2$ [..[7] ]assimilation capacity increases with light, while COS consumption capacity does not. Superimposed upon the
15   light response is a secondary effect [..[8] ]that high vapor deficit further reduces LRU, causing LRU minima to occur in the afternoon, not at noon. The partial stomatal closure [..[9] ]induced by high vapor deficit suppresses COS uptake more strongly than CO$_2$ uptake because stomatal resistance is a more dominant component in the [..[10] ]total resistance of COS. Using stomatal conductance estimates, we show that LRU variability can be explained in terms of different patterns of
* * *
[1]removed: vary with

[2]removed: the LRU versus light relationship

[3]removed: ,

[4]removed: COS and CO$_2$

[5]removed: the mean value reported from laboratory measurements

[6]removed: pressure deficit(or evaporative demand)

[7]removed: reactions are lightlimited but the COS reaction is not. In

[8]removed: , high evaporative demand tends to reduce LRUvalues. During periods of high evaporative demand, leaves conserve water by

[9]removed: . This reduces

[revised manuscript text omitted]

[71]removed: ($F_{COS}$ and $F_{CO_2}$)

[72]removed: concentrations ($\chi_{COS}$ and $\chi_{CO_2}$),

We also calculate the all-day mean LRU ($\mathrm{LRU}_{\text{all-day}}$) and the daytime mean LRU ($\mathrm{LRU}_{\text{daytime}}$) of each day using

$$\mathrm{LRU}_{\text{all-day}} = \frac{\left(\sum\limits_{i=0}^{23} F^i_{\mathrm{COS}}\right) \cdot \left(\sum\limits_{i=0}^{23} \chi^i_{\mathrm{CO_2}}\right)}{\left(\sum\limits_{i=0}^{23} F^i_{\mathrm{CO_2}}\right) \cdot \left(\sum\limits_{i=0}^{23} \chi^i_{\mathrm{COS}}\right)} \tag{10}$$

$$\mathrm{LRU}_{\text{daytime}} = \frac{\left(\sum\limits_{i=6}^{19} F^i_{\mathrm{COS}}\right) \cdot \left(\sum\limits_{i=6}^{19} \chi^i_{\mathrm{CO_2}}\right)}{\left(\sum\limits_{i=6}^{19} F^i_{\mathrm{CO_2}}\right) \cdot \left(\sum\limits_{i=6}^{19} \chi^i_{\mathrm{COS}}\right)} \tag{11}$$

where $i$ is the truncated hour number (integer), in local daylight-saving time (UTC–7). The daytime period is determined with solar elevation angle > 0°, which translates roughly to between 06:00 and 20:00. In each period of calculation, missing data points are gap-filled with the mean in that period.

**2.5.3 Contributions of stomatal component to the total resistance**

To assess the relative importance of the stomatal limitation on COS and $CO_2$ uptake with respect to internal limitations (mesophyll conductance and biochemical reactions), we calculate the ratios of stomatal resistance to total resistance for COS ($r^*_{\mathrm{COS}}$) and $CO_2$ ($r^*_{\mathrm{CO_2}}$),

$$r^*_{\mathrm{COS}} = \frac{r_{s,\,\mathrm{COS}}}{r_{\mathrm{tot},\,\mathrm{COS}}} = \frac{g_{\mathrm{tot},\,\mathrm{COS}}}{g_{s,\,\mathrm{COS}}} = \frac{g_{\mathrm{tot},\,\mathrm{COS}}}{g_{s,\,\mathrm{H_2O}}/2.01} \tag{12}$$

$$r^*_{\mathrm{CO_2}} = \frac{r_{s,\,\mathrm{CO_2}}}{r_{\mathrm{tot},\,\mathrm{CO_2}}} = \frac{g_{\mathrm{tot},\,\mathrm{CO_2}}}{g_{s,\,\mathrm{CO_2}}} = \frac{g_{\mathrm{tot},\,\mathrm{CO_2}}}{g_{s,\,\mathrm{H_2O}}/1.66} \
[revised manuscript text omitted]

[120] removed: 5 ± 1

[121] removed: nocturnal stomatal conductance to COS ($g_{\text{s, COS}}$) if internal conductance ($g_{\text{i, COS}}$), the combination of mesophyll conductance and biochemical reaction coefficient, is ignored ($g_{\text{s, COS}} \ll g_{\text{i, COS}}$). This translates to 10 ± 2

[122] removed: for the stomatal conductance to water ($g_{\text{s}}$), after accounting for the different diffusivities of water and COS in the air with a ratio of 2.0 (Seibt et al., 2010). The nocturnal $g_{\text{s, COS}}$

[123] removed: for other ecosystems, ranging from

[124] removed: to 5–20

[125] removed: Although these observations span a wide range of values across plant species and ecosystem types, the fraction of nocturnal uptake in the daily canopy COS budget lies in a much narrower range of 17–30%. This convergence indicates that nocturnal values may be directly coupled to daytime stomatal conductance . Hence, it may be beneficial for large scale applications to relate nocturnal stomatal conductance to daytime observable parameters, e.

g., 5.5% of the light saturated value for a wheat field (Maseyk et al., 2014) or 2.5% of the daytime maximum value in a New England mixed forest (Wehr et al., 2017).

[126] removed: nocturnal stomatal conductance has been typically parameterized with a small

[127] removed: , for example,

[128] removed: This

[129] removed: in

[130] removed: For better estimates of nighttime COS fluxes and transpiration,

[131] removed: nocturnal stomatal conductance , and its links to daytime values, need to be quantified across

10  ecosystem types. [..[132] ]Water and COS flux measurements need to be used in conjunction to derive robust estimates of nighttime stomatal conductance. We expect COS measurements to be particularly [..[133] ]useful for stomatal conductance estimates in tropical rainforests and other environments that experience high humidity conditions, provided that the variability of the internal conductance of COS is well understood.

**4.3  [..[134] ][..[135] ]Implications on COS-based GPP estimation**

LRU is an important [..[136] ]empirical parameter used to derive GPP from COS measurements on spatial scales ranging from the ecosystem to the continent (Asaf et al., 2013; Commane et al., 2015; Hilton et al., 2015). Choosing a represen-
5  tative LRU for COS-based GPP estimation is crucial and challenging.

[..[137] ][..[138] ][..[139] ]

[..[140] ]
* * *
[132]removed: COS measurements are well suited for this purpose since COS uptake continues as long as stomata are open, whereas water fluxes become very small as the ambient air typically gets close to saturation at night

[133]removed: beneficial

[134]removed: The environmental determinants of leaf relative uptake (LRU)

[135]removed: Leaf COS to $CO_2$ relative uptake (LRU )

[136]removed: parameter that links plant COS uptake with GPP . Observations at leaf and ecosystem scales show that LRU is primarily controlled by light, following an asymptotically decreasing trend with increasing PAR (Fig. 5c; Stimler et al., 2010, 2011; Maseyk et al., 2014; Commane et al., 2015). Such a pattern originates from the differential responses of COS and $CO_2$ uptake to light, because unlike photosynthesis, COS uptake responds only indirectly to light through changes in stomatal conductance (Stimler et al., 2011). Using the nonparametric LOWESS fit without assuming an a priori relationship between LRU and PAR, we found an LRU–PAR relationship (Fig. 5c) similar to the decaying power law (LRU $= a \cdot \mathrm{PAR}^{-b}$) reported by Maseyk et al. (2014). Based on this and previous studies, the light response of LRU may be generalized empirically with a decaying power law fit (Stimler et al., 2010, 2011; Maseyk et al., 2014; Commane et al., 2015).

[137]removed: We identified vapor deficit as secondary environmental driver of LRU, resulting from the differential effects of low humidity induced stomatal closure on COS and $CO_2$ fluxes (Fig. 5a, b; see also sect. 4.1). High vapor deficit tends to reduce LRU values in mid-afternoon, when LRU is expected to reach light-saturated values according to the LRU–PAR relationship. This is because stomatal conductance is a more dominant component in the diffusional pathway for COS than for $CO_2$. Using the resistance analog (the inverse of conductance, i.e., $r_s = g_s^{-1}$), we can combine all sub-stomatal terms (mesophyll and chloroplast wall conductances and biochemical reaction coefficient) into a single internal resistance term ($r_{i,COS}$ or $r_{i,CO_2}$). Because of the strong affinity of $\beta$-CA for COS (Ogawa et al., 2013), COS is more readily consumed at the CA active site than $CO_2$ is at the carboxylation site of RuBisCO (Stimler et al., 2010; Berry et al., 2013), leading to a much smaller contribution of internal resistance to the COS diffusional pathway,

[138]removed: $\dfrac{r_{i,COS}}{r_{s,COS} + r_{i,COS}} < \dfrac{r_{i,CO_2}}{r_{s,CO_2} + r_{i,CO_2}}$

[139]removed: For example, based on rough estimates of light-saturated values of stomatal conductance ($g_{s,H_2O} = 80$ mmol m$^{-2}$ s$^{-1}$) and COS and $CO_2$ fluxes (Fig.

5), for a relative decrease in stomatal conductance ($g_{s,COS}$ and $g_{s,CO_2}$) of 50% at high vapor deficit, the total resistance of COS uptake increases by 37% whereas that of $CO_2$ uptake only increases by 28%. Thus, when $CO_2$ uptake is light saturated, a decrease in stomatal conductance due to high vapor deficit will reduce COS uptake more than $CO_2$ uptake, and result in a lower LRU (7% for the examples above, from 1.07 to 1.0).

[140]removed: Previous laboratory studies have not found any significant response of LRU to relative humidity (Stimler et al., 2010, 2011), but it is possible that the vapor deficit in the experiments was not strong enough to initiate partial stomatal closure. At our site, the influence from vapor deficit causes the lowest LRU values to occur in the early afternoon (15:00), when vapor deficit is the highest, instead of at noon when PAR is highest (13:00).

[revised manuscript text omitted]

**5  Conclusions**

[..[156] ]Our field study has shown that leaf COS and $CO_2$ fluxes share [..[157] ]similar diurnal patterns driven by the common stomatal responses to light and vapor deficit[..[158] ], showing dual peaks of uptake separated by a prolonged midday depression period. We have validated the light dependence of LRU directly at the leaf level in field conditions. LRU converges to around 1.0 at light-saturated conditions for *Typha latifolia*, much lower than many other species due possibly to its high photosynthetic capacity. In addition to light, vapor deficit is identified as a secondary driver of LRU, acting to reduce LRU further in the afternoon (15:00–17:00) from its light-saturated value.

[..[159] ][..[160] ]Stomatal conductance derived from water measurements has provided process-level insights into the diurnal variability of LRU. Since the biochemical sink of COS is light independent, COS uptake is less reaction-limited compared with $CO_2$ uptake. With increasing light, the assimilation capacity for $CO_2$ increases but is unchanged for COS, causing LRU to [..[161] ][..[162] ][..[163] ]decrease regardless of the stomatal coupling between COS and $CO_2$. The reduction
* * *
[153]removed: We found a good correlation between daytime mean LRUand daytime mean PAR ($r = -0.525$;

[154]removed: 6b), similar to Maseyk et al. (2014). This indicates that

[155]removed: LRU–PAR relationship is preserved at the daily timescale, supporting the use of COS as a photosynthetic tracer at large scales where measurements are often made at daily or longer intervals. On overcast days, the daytime mean LRU values were higher than on clear days (Fig. 6a), as expected from the light response of LRU . We expect the relationship between daytime means of LRU and PAR to be useful for calculating daytime mean LRU empirically from meteorological conditions for GPP estimates. Since the use of COS as a GPP tracer in an inverse modeling framework requires the uncertainty in LRU to be smaller than that in the a priori GPP estimates (Hilton et al., 2015), future studies should be dedicated to understanding LRU variability in the field for accurate COS-based GPP estimates

[156]removed: From direct field observations at the leaf scale, our

[157]removed: broadly

[158]removed: . In the early morning and late afternoon, the increase of COS uptake with light is caused by increasing stomatal conductance, since the COS reaction with CA is light independent. Around midday, vapor deficit becomes a limiting factor of stomatal conductance and drives the midday depression in COS and $CO_2$

[159]removed: We have identified three distinct physiological regimes that control LRU variability over the course of a day:

[160]removed: In the early morning when both PAR and vapor deficit are low, biochemical reactions of $CO_2$ are light limited. As a result, leaf $CO_2$ uptake is more restricted than COSuptake that is only stomatal conductance limited

[161]removed: be high and to decrease with PAR.

[revised manuscript text omitted]

---

## Referee Report (RR1)

General comments:

This manuscript describes the efforts to characterize the leaf relative uptake (LRU) under natural field conditions. Understanding the variability of this parameter is necessary to link COS fluxes to gross primary production. This study is carried out adequately with a thorough analysis and interpretation of the available data and it contributes to the understanding of the variability of LRU.

The manuscript has improved now that it is shown with data that the share of stomatal resistance to the total resistance is larger for COS than for $CO_2$. This provides evidence that COS is indeed more stomatal limited than $CO_2$, which was hypothesized, but not shown with data in the previous version of the manuscript. The main concern that I have is that the second hypothesis in the introduction is not well introduced. The introduction describes the expected light dependence of LRU well (hypothesis 1), but the hypothesis that diurnal variation of vapor deficit will have effects on LRU is not explained here at all. This deserves some explanation in the introduction already.

Specific comments:

Introduction

Page 3, line 6-7: reference missing.

Page 3, line 17-18: Introduce the hypothesis that LRU will depend on the diurnal variation of vapor deficit.

Results

At the end of each results section (3.1, 3.2, 3.3) there is an interpretation of the data that I think would fit better in the discussion section: page 8, line 32-33; page 9, line 11-14; page 9, line 32.

Page 9, line 12-13: "For COS, stomatal limitation is always a much stronger component compared with that of $CO_2$." Rather say how much the difference is on average, instead of stating "much stronger".

Page 9, line 22: "[…] due to the stronger stomatal limitation on fluxes as a response to the high vapor deficit." It has not been introduced here why stomatal limitation would affect LRU. Such interpretation would fit better in the discussion section, and it would have to be explained (preferably already in the introduction) why/how the stomatal conductance affects the LRU.

Discussion

Page 10, line 11-13: "This light response of LRU arises from the difference between the marginal gain (i.e., partial derivative) of COS uptake and that of $CO_2$ uptake with respect to the

same increase of PAR (Fig. 5a, b)." It is not clear to me what you mean here, can you describe it in other words?

Page 10, line 16-19: This is not easy to follow. Perhaps it is easier to comprehend if you explain it in terms of Fcos and Fco2 (Fig 5a-b?) than in terms of $r_{cos}$ and $r_{co2}$? Also I do not find it that evident in Fig. 6b that the relative increase of $r_{co2}$ is higher than that of $r_{cos}$, it would be helpful if you can provide numbers of the relative increase of each.

Page 10, line 28-29: If you want to introduce the hypothesis that LRU depends on vapor deficit in the introduction section then it would be good to mention the difference between the catalytic efficiencies there already.

Supplement

S1: "For COS, the use of a correction factor of 1.0 was acceptable."

This is only in the case that the instrument software fitting parameters split the fit between the COS and $H_2O$ peak, so that the $H_2O$ peak does no longer influence the COS peak. Was that the case? If not, the correction factors -0.0146 (for $CO_2$) and 0.030 (for COS), e.g. $[CO_2dry]$ = $[CO_2wet]/(corr.fact.*[H_2O]+1)$ suggested by Kooijmans et al. (2016) should be used.

---

## Author Response (AR2)

**Response to the comments on the revised version of "Stomatal control of leaf fluxes of carbonyl sulfide and CO$_2$ in a *Typha* freshwater marsh"**

Wu Sun, on behalf of all authors

In this response, the text is formatted as: referees' comments (indented blocks), authors' response, **applied changes in the manuscript**. Locations in the revised manuscript are referenced by "P{#X}L{#Y}", meaning "Line Y on Page X".

**Reply to comments by Teresa E. Gimeno (Referee #1)**

> This is the second time I review this manuscript and I would like to highlight again the clarity of the text and the presentation of the results. I believe this manuscript has improved significantly with respect to its previous version. The theoretical framing of the study is more solid, I really appreciated the author's effort to formulate two specific hypotheses. I have some comments on that regard that I detail below. The technical clarifications added are pertinent and specific and I only have a few minor questions (again see below). I think the addition of the detailed explanation of the calculations underlying the different diffusion components and the inclusion of the stomatal conductance was needed. The discussion has vastly improved with respect to the previous version, being a lot more interesting and better suited to the results presented.

We'd like to thank Dr. Gimeno again for her time and effort devoted to the evaluation of the manuscript. We appreciate her helpful comments that have improved our manuscript. Below we address the new concerns.

> My biggest concern is still related to the limited scope of the paper. The authors need to acknowledge the limitations of their study clearly. As they stated in the response letter themselves, replication is always desirable in any study and although any potential reader (as well I myself or the handling editor) would recognise the technical and practical limitations of any field study, these limitations need to be stated clearly. The main limitation of this study is that only one set of six leaves (from the same plant?) was measured continuously and that the target species is a bit particular in terms of physiological behaviour. This study focuses on a salt-marsh plant, which undoubtedly serves as an excellent target for a proof-of-concept type study. However, it should be noted that the observed results might be influenced by its particular physiology and habitat preference, as noted on the comments to the previous version by Dr. M. E. Whelan and myself. Besides, it appears that these leaves might not necessarily be the most representative as "the chamber received slightly more light in the afternoon than in the morning due to a wider gap in the canopy to the west of the chamber than to other directions" (P10L9). I am not saying that these circumstances ought to be a weakness, but $CO_2$ and COS patterns could have coupled differently under different circumstances, and this is an important limitation that needs to be acknowledged very early in the discussion.

These limitations have now been acknowledged more explicitly. **We have clarified that the leaves were on the same plant (P4L5 in §2.2). Limitations regarding the lack of replication and the heterogeneity of the chamber light environment have been acknowledged in the Methods (P4L8–9 and P5L5–7 in §2.2).**

Furthermore, despite the limitations in our study, we expect the behavior to be a general characteristic. We have observed similar behavior in at least one other species, Scots pine, in a later study (manuscript submitted two weeks ago by Linda Kooijmans et al.).

All my comments below refer to the page and line numbers (PXLY) on the manuscript version with the track changes where insertions and deletions with respect to the previous version were indicated.

P3L11: Please correct me if I am wrong, but I believe that there is not "only one study reporting nighttime COS uptake at the leaf scale" since besides Berkelhammer et al. (*Global Biogeochemical Cycles*, 28, 161–179, 2014), Kooijmans et al. (*Atmos. Chem. Phys.*, 17, 11453–11465, 2017) also report chamber-level measurements of nocturnal COS uptake in the field and Stimler et al. (*New Phytologist*, 186, 869–878, 2010) report leaf level COS uptake under laboratory conditions in the dark. I believe the authors might want to consider rephrasing this statement and/or adding some citations.

We agree with the reviewer on the contributions of these works. There were some confusions about this statement. We intended to mean that there had only been one study (Berkelhammer et al., 2014) measuring nighttime COS uptake *both* at the leaf scale *and* in the field. Precisely,

the study of Stimler et al. (2010) was done in a laboratory setting, while Kooijmans et al. (2017) used ecosystem COS uptake *and* independent estimates of stomatal conductance (measured at the leaf scale) to reason about the nighttime canopy COS uptake. But by no means did we intend to omit their contributions. In fact, all of them were cited earlier in the same paragraph. **We have reworked this sentence to eliminate potential confusions, which now reads: "Most studies base their findings of nighttime COS uptake upon ecosystem scale observations, with only a handful of studies providing leaf-level evidence of nighttime COS uptake (Stimler et al., 2010; Berkelhammer et al., 2014; Kooijmans et al., 2017)."** See P2L29–31.

> P3L20–P4L9: In my opinion, and in line with the previous comments raised by Dr. M. E. Whelan, I believe the manuscript would benefit greatly from reducing the emphasis on the relevance of LRU. This paragraph provides a nice review of previously reported responses of LRU to environmental drivers, but it would be more appropriate to reduce it to one or two sentences targeted at formulating the study hypotheses at the relevant scale (leaf).

**This paragraph has now been merged into the preceding paragraph to better serve the introduction of the hypotheses.** It was not possible to cut it down to two sentences, because we have added some elaboration on the possible vapor deficit dependence of LRU per request of Reviewer #2. We have aimed to keep the whole paragraph concise without losing clarity.

> P4L12: I really appreciated the author's effort to formulate specific hypotheses; nevertheless, I do not follow what information given on the introduction leads to formulating hypothesis (ii). I think this needs to be clarified.

**This hypothesis has been reframed for clarity.** See P3L14–15.

> P5L10: How was leaf area 'estimated'?

The area of each leaf was approximated with a one-sided rectangle (i.e., length intersected by the chamber × width). This was because the leaves were vertically oriented and were bundled together such that only one side was exposed for gas exchange. **We have described this in P4L6–7.**

> Remove "reaffirming the shared stomatal control on both fluxes" (P11L23-25, on the top), also "This indicates that stomatal conductance exerted a stronger control on

COS uptake than $CO_2$ uptake" (P11L6-7) and "This difference may be attributed to changes in internal conductance terms entailed in $g_{tot}$, COS, namely, mesophyll conductance and biochemical activities." (P11L16-17). Speculations and suggestions on which processes underlie these correlations and patterns belongs to the discussion.

**These sentences have been removed from the Results at the requests of both reviewers.**

P11L20: I am not sure I understand how Figure 6b illustrates how stomata opening limited COS diffusion more strongly than for $CO_2$ since the 25–75th percentile intervals appear to overlap completely for these two gases. I would like to ask the authors to consider rephrasing or even removing this statement, or maybe to accompany with some sort of formal statistical test.

**Uncertainties associated with the data have been addressed in P9L11–15 in §3.2. We have also added markers to indicated significance levels derived from the paired two-sample *t*-test in Fig. 6b for the comparisons.**

P12L12-13: In agreement with my previous comments, I consider that this ["when the difference between stomatal limitation..."] could be viewed as an over-interpretation of the observed patterns.

**This sentence has been removed from the Results section.**

P12L19: Could you please provide the P-value for this correlation?

**The *p*-value have been provided in P9L31 in §3.3.**

P12L21-22: I would move the last part of this paragraph ["similar to (. . .) the daily timescale"] to the discussion.

**Removed.**

P12L24: I agree that stomatal and other internal diffusion resistances (which are not accounted for here) are likely to underlie the observed patters of COS and $CO_2$ uptake measured here. Nevertheless, maybe the authors should consider opening their discussion by briefly reminding the reader whether their results agree with their original hypotheses. In addition, I think it is very important to state here, very clearly, the limitations of the study, mainly that observations are limited to one single set of leaves in a salt marsh plant, which has a very particular physiology. In the previous review, both Dr. M. E. Whelan and myself noted several particularities of the physiology of this plant and these need to be incorporated in the discussion as they might influence the coupling of COS and $CO_2$ fluxes and their underlying regulation by stomatal conductance.

**We have rewritten the opening paragraph of the §4.1 to remind the audience about the hypotheses and also to state the limitations of the study, particularly the lack of replication. The limitation associated with the unique physiology of the marsh plant has been acknowledged in P12L8–12 in §4.3.**

Figure 4. I understand that the correlation coefficient ($r^2$) from figure (a) corresponds to a linear fit between leaf $CO_2$ and COS fluxes, but I cannot see what would be the equivalent fit in (b). This needs to be clarified in the figure legend. Also, please consider providing P-values here too.

It is true that Pearson's correlation coefficient as a measure of dependence is biased by nonlinearity. In light of this issue, **we have also added the distance correlation (dCor) that suits the testing of nonlinear dependence.** See P8L26–27 in §3.1 and annotations on Fig. 4b.

$p$-values corresponding to the Pearson's correlation coefficients are not shown on the figure because they are smaller than the machine epsilon of double-precision floating point number ($\sim 1 \times 10^{-16}$), but they have nevertheless been provided in P8L23 and P8L25 in §3.1.

**Reply to comments by Referee #2**

**General comments**

This manuscript describes the efforts to characterize the leaf relative uptake (LRU) under natural field conditions. Understanding the variability of this parameter is necessary to link COS fluxes to gross primary production. This study is carried out adequately with a thorough analysis and interpretation of the available data and it

contributes to the understanding of the variability of LRU.

We thank the reviewer's evaluation of our manuscript.

The manuscript has improved now that it is shown with data that the share of stomatal resistance to the total resistance is larger for COS than for $CO_2$. This provides evidence that COS is indeed more stomatal limited than $CO_2$, which was hypothesized, but not shown with data in the previous version of the manuscript. The main concern that I have is that the second hypothesis in the introduction is not well introduced. The introduction describes the expected light dependence of LRU well (hypothesis 1), but the hypothesis that diurnal variation of vapor deficit will have effects on LRU is not explained here at all. This deserves some explanation in the introduction already.

**We have now rewritten part of the introduction that leads to the second hypothesis. The hypothesis itself has also been rephrased for clarity.**

**Specific comments**

**Introduction**

Page 3, line 6-7: reference missing.

**This sentence has been rephrased, and references have been added.** See P3L3–4.

Page 3, line 17-18: Introduce the hypothesis that LRU will depend on the diurnal variation of vapor deficit.

**This hypothesis has been reframed for clarity.** See P3L14–15. A priori information that leads to the formulation of this hypothesis has also been added in previous paragraphs.

**Results**

At the end of each results section (3.1, 3.2, 3.3) there is an interpretation of the data that I think would fit better in the discussion section: page 8, line 32–33; page 9, line 11–14; page 9, line 32.

**These 'interpretations' have been removed from the Results and assimilated into the Discussion.**

Page 9, line 12–13: "For COS, stomatal limitation is always a much stronger component compared with that of $CO_2$." Rather say how much the difference is on average, instead of stating "much stronger".

**This sentence has been revised to incorporate quantitative information and statistical significance.** See P9L11–13.

Page 9, line 22: "[. . .] due to the stronger stomatal limitation on fluxes as a response to the high vapor deficit." It has not been introduced here why stomatal limitation would affect LRU. Such interpretation would fit better in the discussion section, and it would have to be explained (preferably already in the introduction) why/how the stomatal conductance affects the LRU.

Agreed. **This sentence has been removed from the Results. In addition, the background knowledge that leads to this interpretation has been added in the Introduction.** See P2L21 and P3L6–8.

**Discussion**

Page 10, line 11–13: "This light response of LRU arises from the difference between the marginal gain (i.e., partial derivative) of COS uptake and that of $CO_2$ uptake with respect to the same increase of PAR (Fig. 5a, b)." It is not clear to me what you mean here, can you describe it in other words?

**This has been clarified in P10L15–19.**

Page 10, line 16–19: This is not easy to follow. Perhaps it is easier to comprehend if you explain it in terms of Fcos and Fco2 (Fig 5a–b?) than in terms of rcos and rco2? Also I do not find it that evident in Fig. 6b that the relative increase of rco2 is higher than that of rcos, it would be helpful if you can provide numbers of the relative increase of each.

**We have clarified these explanations.** However, both explanations have been kept because they provide multiple lines of evidence to support the mechanism. See P10L19–24.

Page 10, line 28-29: If you want to introduce the hypothesis that LRU depends on vapor deficit in the introduction section then it would be good to mention the difference between the catalytic efficiencies there already.

Agreed. **Moved to P2L15–17 in the Introduction.**

**Supplement**

S1: "For COS, the use of a correction factor of 1.0 was acceptable."

This is only in the case that the instrument software fitting parameters split the fit between the COS and $H_2O$ peak, so that the $H_2O$ peak does no longer influence the COS peak. Was that the case? If not, the correction factors −0.0146 (for $CO_2$) and 0.030 (for COS), e.g. [$CO_{2 \, dry}$] = [$CO_{2 \, wet}$] / (corr. fact. * [$H_2O$] + 1) suggested by Kooijmans et al. (2016) should be used.

The broadening coefficients of 0.030 for COS and −0.0146 for $CO_2$ apply to the "standard fit, water correction off" case, which was not our case. We used "standard fit" but always had the water vapor correction option turned *on*. This was the setting recommended by the manufacturer back at the time when the fieldwork was carried out (which preceded the study of Kooijmans et al., 2016).

In Kooijmans et al. (2016), the broadening coefficient of COS was not given for the "standard fit, water correction on" case because it can range from 1.0 to 1.5. However, they also noted that this uncertainty has relatively little influence on COS concentration:

"We find that the uncertainties of the broadening coefficients are equal to 0.5 (COS), 0.03 ($CO_2$) and 0.7 (CO). This means that varying the broadening coefficient of COS from 1.0 to 1.5 only changes the COS concentration by 2.9 ppt (at a concentration of 450 ppt COS)."

Therefore our choice of 1.0 was acceptable, although it was at the lower end of the possible range of COS broadening coefficients.

Please also note that the broadening coefficients for the "off" cases in Kooijmans et al. (2016) are given with respect to $H_2O$ concentration in *percentage*, whereas the default broadening coefficients provided by the manufacturer for the "on" cases are given for $H_2O$ concentration in *molar fraction* (we used the latter one). If the broadening coefficients in both "on" and "off" cases were to be converted to the same unit of $H_2O$ concentration, they should be on the same order of magnitude.

**Stomatal control of leaf fluxes of carbonyl sulfide and $CO_2$ in a *Typha* freshwater marsh**

Wu Sun[1], Kadmiel Maseyk[2,a], Céline Lett[3,a], and Ulli Seibt[1,a]

[1]Department of Atmospheric and Oceanic Sciences, University of California, Los Angeles, CA 90095-1565, USA
[2]School of Environment, Earth and Ecosystem Sciences, The Open University, Milton Keynes MK7 6AA, United Kingdom
[3]Laboratoire des Sciences du Climat et de l'Environnement, Université Paris Saclay, 91191 Gif-sur-Yvette, France
[a]formerly at Institute of Ecology and Environmental Sciences, Université Pierre et Marie Curie Paris 6, France

**Correspondence:** Wu Sun (wu.sun@ucla.edu) and Ulli Seibt (useibt@ucla.edu)

**Abstract.** Carbonyl sulfide (COS) is an emerging tracer to constrain land photosynthesis at canopy to global scales, because leaf COS and $CO_2$ uptake processes are linked through stomatal diffusion. The COS tracer approach requires knowledge of the concentration normalized ratio of COS uptake to photosynthesis, commonly known as the leaf relative uptake (LRU). LRU is known to increase under low light, but the environmental controls over LRU variability in the field are poorly understood due to scant leaf scale observations.

   Here we present the first direct observations of LRU responses to environmental variables in the field. We measured leaf COS and $CO_2$ fluxes at a freshwater marsh in summer 2013. Daytime leaf COS and $CO_2$ uptake showed similar peaks in the mid-morning and late afternoon separated by a prolonged midday depression, highlighting the common stomatal control on diffusion. At night, in contrast to $CO_2$, COS uptake continued, indicating partially open stomata. LRU ratios showed a clear relationship with photosynthetically active radiation (PAR), converging to 1.0 at high PAR, while increasing sharply at low PAR. Daytime integrated LRU (calculated from daytime mean COS and $CO_2$ uptake) ranged from 1 to 1.5, with a mean of 1.2 across the campaign, significantly lower than previously reported laboratory mean value (~1.6). Our results indicate two major determinants of LRU—light and vapor deficit. Light is the primary driver of LRU because $CO_2$ assimilation capacity increases with light, while COS consumption capacity does not. Superimposed upon the light response is a secondary effect that high vapor deficit further reduces LRU, causing LRU minima to occur in the afternoon, not at noon. The partial stomatal closure induced by high vapor deficit suppresses COS uptake more strongly than $CO_2$ uptake because stomatal resistance is a more dominant component in the total resistance of COS. Using stomatal conductance estimates, we show that LRU variability can be explained in terms of different patterns of stomatal vs.[..[1] ] internal limitations on COS and $CO_2$ uptake. Our findings illustrate the stomata-driven coupling of COS and $CO_2$ uptake during the most photosynthetically active period in the field and provide an in-situ characterization of LRU—a key parameter required for the use of COS as a photosynthetic tracer.

*Copyright statement.* © 2018 Authors. This work is licensed under a Creative Commons Attribution 4.0 International License (CC BY 4.0).
* * *
[1]removed:

**1 Introduction**

[revised manuscript text omitted]